# The shifting of secondary inorganic aerosols formation mechanism during haze aggravation: The decisive role of aerosol liquid water

Fei Xie[1, 2], Yue Su[1, 3], Yongli Tian[2], Yanju Shi[2], Xingjun Zhou[2], Peng Wang[2], Ruihong Yu[1], Wei Wang[1], Jiang He[1, 3], Jinyuan Xin[4, *], Changwei Lü[1, 3, *]

[1] *School of Ecology and Environment, Inner Mongolia University, 010021, Hohhot, China*

[2] *Inner Mongolia Environmental Monitoring Center, 010011, Hohhot, China*

[3] *Institute of Environmental Geology, Inner Mongolia University, 010021, Hohhot, China*

[4] *State Key Laboratory of Atmospheric Boundary Layer Physics and Atmospheric Chemistry (LAPC), Institute of Atmospheric Physics, Chinese Academy of Sciences, Beijing 100029, China*

## Abstract

Although many considerable efforts have been done to reveal the driving factors on haze aggravation, however, the roles of aerosol liquid water (ALW) in SIAs formation were mainly focused on the condition of aerosol liquid water content (ALWC)<100 $\mu g/m^3$. Based on the in-situ high-resolution field observation, this work studied the decisive roles and the shifting of secondary inorganic aerosols formation mechanism during haze aggravation, revealing the different roles of ALWC in a broader scale ($\sim$ 500 $\mu g/m^3$) in nitrate and sulfate formation induced by aqueous chemistry in ammonia-rich atmosphere. The results showed that chemical domains of perturbation gas limiting the generation of secondary particulate matters presented obvious shifts from $HNO_3$ sensitive to $HNO_3$ and $NH_3$ co-sensitive regime with the haze aggravation, indicating the powerful driving effects of ammonia in ammonia-rich atmosphere. When ALWC<75 $\mu g/m^3$, the sulfate generation was preferentially triggered by the high ammonia utilization, then accelerated by nitrogen oxide oxidation from Clean to Moderate pollution stages, characterizing as nitrogen oxidation ratio (NOR)<0.3, sulfur oxidation ratio (SOR)<0.4, ammonia transition ratio (NTR)<0.7 and the moral ratio of $NO_3^-/SO_4^{2-}$=2:1. While ALWC>75 $\mu g/m^3$, aqueous-phase chemistry reaction of $SO_2$ and $NH_3$ in ALW became the prerequisite for SIAs formation driven by Henry's law in the ammonia-rich atmosphere during Heavy and Serious stages, characterizing as high SOR (0.5-0.9), NOR (0.3-0.5), NTR (>0.7) and the moral ratio of $NO_3^-/SO_4^{2-}$=1:1. A positive feedback of sulfate on nitrate production was also observed in this work due to the shift of ammonia partition induced by the ALWC variation during haze aggravation. It implies the target controlling of haze should not simply focus on $SO_2$ and $NO_2$, more attention should be paid on gaseous precursors (e.g., $SO_2$, $NO_2$, $NH_3$) and aerosol chemical constitution during different haze stages.

**Keywords:** Mechanism shifting, Aerosol liquid water, Secondary inorganic aerosols, Haze aggravation, In-situ observation

---

* Corresponding author, Email: xjy@mail.iap.ac.cn; lcw2008@imu.edu.cn

## 1 Introduction

Fine particulate matter (PM$_{2.5}$) presented close link with several environmental issues, such as visibility reduction and climate change (Zhang et al., 2015; Shang et al., 2020; Wang et al., 2020; Wang et al., 2016; Nozière et al., 2010). Epidemiological studies have stated the association of PMs with various public health, even adverse birth outcomes (Gwynn et al., 2000; Lavigne et al., 2016; Zhao et al., 2020). As the most abundant secondary inorganic aerosols (SIAs) in PM$_{2.5}$ during Chinese winter haze episodes (Fu and Chen, 2017; Liu et al., 2019), the formations of sulfate and nitrate play the key roles during haze aggravation, as well as the impacting factors of the oxidants in gas and aqueous phases, the characteristics of pre-existing aerosols/fog/cloud, and meteorological conditions. Recently, aerosol liquid water content (ALWC) was reported associating with the SIAs formation, especially sulfates and nitrates, during the haze periods (Wu et al., 2018; Zheng et al., 2015a; Wang et al., 2016; Cheng et al., 2016; Carlton and Turpin, 2013; Nguyen et al., 2014; Xue et al., 2014; Tan et al., 2017; Liu et al., 2017b). Atmospheric aerosol liquid water (ALW), which determined by ambient relative humidity (RH), has been proposed as a container since it could provide the reaction medium for the multiphase chemistry during the haze process (Ansari and Pandis, 2000; Shiraiwa et al., 2012; Davies and Wilson, 2015). The roles of ALWC on the generations of particulate sulfate generations (Wang et al., 2016; Cheng et al., 2016) and global secondary organic aerosols (Hodas et al., 2014; Mcneill, 2015; Wong et al., 2015) were reported. Thus, fully understanding ALW and its roles during haze aggravation is fundamentally important on atmospheric physicochemical processes, especially the liquid chemical transformation of SO$_2$ and NO$_x$ in ALW.

Ammonia is the most important alkaline gas, neutralizing with acidic species to form ammonium salts. Due to little attention has been paid to NH$_3$ emissions by Chinese government, atmospheric NH$_3$ experienced a significant increasing trend (Ge et al., 2019; Tan et al., 2017). Although the increase in atmospheric NH$_3$ is beneficial to reduce atmospheric acidity (Liu et al., 2019), its chemical behavior on regional haze formation is still debating. Cheng et al. (2016) indicated that the fast transform of gaseous SO$_2$ to particle sulfate under polluted conditions is attributed to the neutralization of NH$_3$, which raises particle pH and thereby facilitated the aqueous oxidation of S (VI) by NO$_2$. Fang et al. (2017) stated that NH$_3$ partition significantly modified aerosol pH and thereby adjusting the partition of SO$_2$ and NO$_2$. Although the role of NH$_3$ has been identified from a theoretical perspective, the lack of NH$_3$ emission control sets barriers for more effective reduction of PM$_{2.5}$. Therefore, it is urgent to fully understand the reactive gases behavior and the chemical mechanism of SIAs formation during different

pollution stages, which will be helpful to propose reasonable strategies for each stage.
So far, the SIAs formation has been extensively studied during short-term, continuous, or
persistent haze episodes, proposing several heterogeneous and homogeneous oxidation
pathways on sulfate and nitrate formation (Guo et al., 2014; Guo et al., 2017; Zheng et al.,
2015b; Huang et al., 2014; Liu et al., 2021; Yao et al., 2020; Zhou et al., 2018; Liu et al., 2019).
In ammonia-rich atmosphere, $NH_3$ partition significantly modified aerosol pH, adjusted the
partition of $SO_2$ and $NO_2$ (Fang et al., 2017) and promotes the aqueous oxidation of S (VI) by
$NO_2$ (Wang et al., 2016; Cheng et al., 2016). Although many considerable efforts have been
done to reveal the driving factors on haze aggravation, however, the roles of ALW in SIAs
formation were mainly focused on the condition of ALWC<100 $\mu g/m^3$ (Nenes et al., 2020; Wu
et al., 2018; Bian et al., 2014; Jin et al., 2020). Therefore, the roles of ALWC in a broader scale
and the mechanism shifting of secondary inorganic aerosols formation during haze aggravation
in ammonia-rich atmosphere need to be understood in depth. Based on a continuous
observation with 1-hour resolution from December 2019 to January 2020, this work discussed
the shift of dominant mechanism with ALWC variation during the time window of haze
aggravation processes, which will be helpful to propose more effective $PM_{2.5}$ control strategies
for each pollution stage.

## 2 Sampling and Experiment Methods

### 2.1 Description of Sampling Site

Hohhot, the capital city of Inner Mongolia Autonomous Region, is the central city of Hohhot-
Baotou-Ordos group, as well as an important northern China city with a population of more
than 3.126 million and an area of 17224 $km^2$ (Fig. 1). This region is featured as continental
climate with marked seasonality changes, which characterized as long-lasting cold humid
winter and short-time other seasons. Thereby, to survive the cold season, approximately half
year of coal-fired heating events (Oct. 15-the following Apr. 15) were introduced, which
emitting gaseous pollutants as well as PMs around-the-clock. The main industries include
thermal power plants, coal-energy based biochemical industry, dairy industry and
petrochemical industry, etc., which also emit atmospheric pollutants ceaselessly. Thus, high
concentrations of PMs pollution cases dominated the major contamination cases during winter
season (data obtained from Department of Ecology and Environment of Inner Mongolia
Autonomous Region, http://sthjt.nmg.gov.cn/) and gradually emerging as the limiting factor on
regional ambient air quality and human health.
In this study, the observation was conducted at the Inner Mongolia Environmental

Monitoring Center (40°49′22″N, 111°45′2″E) on a top of a sixteen-story building (~40m above the ground level) located at the eastern part of the downtown near the People's Government of Inner Mongolia Autonomous Region near the 2nd ring road from December 1, 2019 to January 31, 2020. Residential and administrative regions were characterized as the major functional domain near the sampling site, with no direct industrial regions nearby.

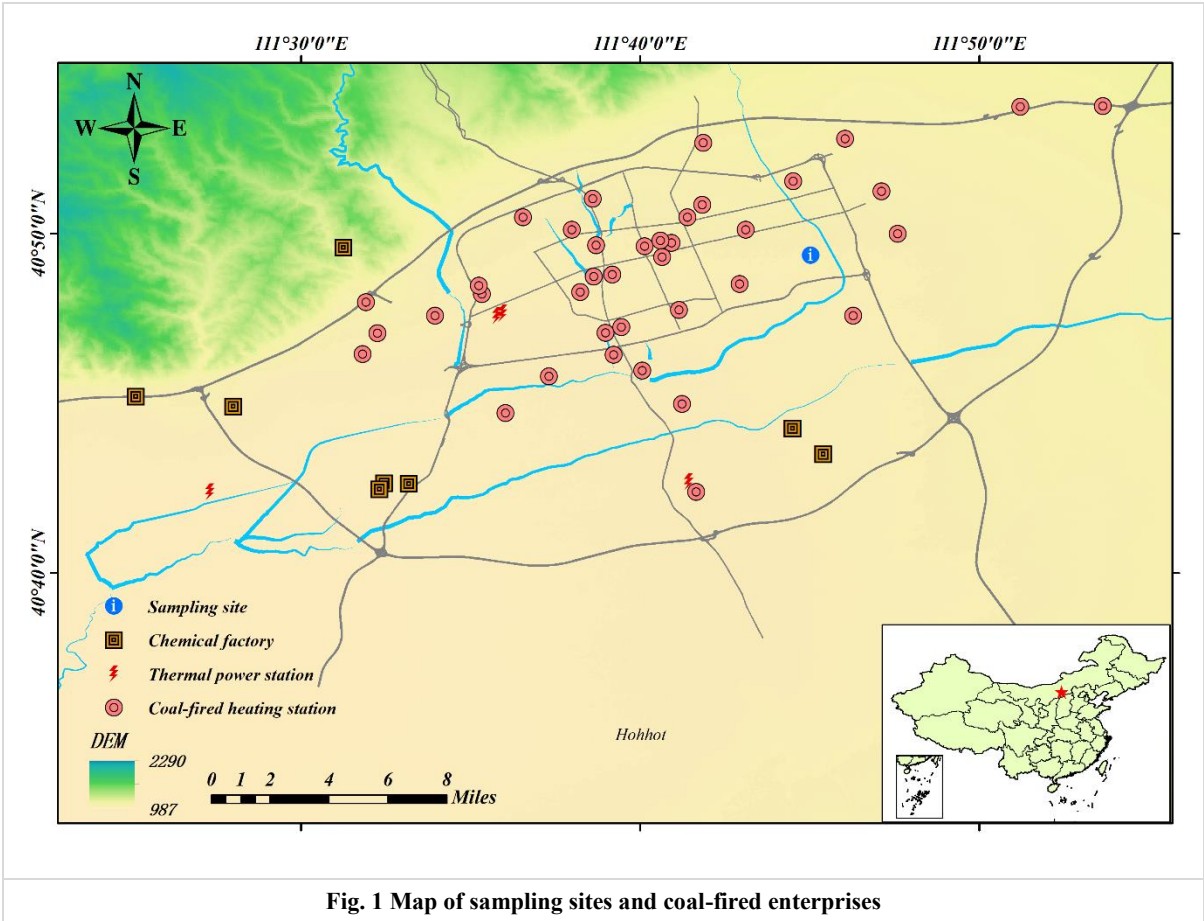

**Fig. 1 Map of sampling sites and coal-fired enterprises**

## 2.2 Data acquisition and analysis methods

### 2.2.1 Data acquisition

On-line ion-chromatograph instrument (MARGA ADI 2080, Metrohm Applikon, Switzerland) was employed to simultaneously determine the water-soluble inorganic ions ($Na^+$, $NH_4^+$, $Mg^{2+}$, $Ca^{2+}$, $K^+$, $Cl^-$, $F^-$, $SO_4^{2-}$, $NO_3^-$) in $PM_{2.5}$ and corresponding trace gases ($SO_2$, $HNO_2$, $HNO_3$, $HCl$, $NH_3$). This instrument has been widely used in previous work (Rumsey et al., 2014; Nie et al., 2015; Huang et al., 2020) and the details were listed in Supplement (S1.1). Correspondingly, gaseous pollutants (e.g., $NO_x$, $CO$, $PM_1$, $PM_{2.5}$, $PM_{10}$) and meteorological datasets (e.g., wind speed, wind direction, RH, temperature, etc.), as well as the adopted models could be found in our previous work (Xie et al., 2021). In addition, peroxyacetyl nitrates (PANs), nitrous oxide ($N_2O$) and solar spectrophotometry were measured by PANs-100 (Focused Photonics Inc.),

$N_2O$ Monitor (LSE, Monitors) and CE-318T (CIMEL), respectively.

***2.2.2 Analysis methods***
Generally, sulfur oxidation ratio (SOR) and nitrogen oxidation ratio (NOR) were calculated as
follows, which were used to indicate the contribution of secondary transformation during the
haze events (Song et al., 2007; Zhou et al., 2018).

$$SOR = \frac{n(SO_4^{2-})}{n(SO_2) + n(SO_4^{2-})}$$


$$NOR = \frac{n(HNO_3) + n(NO_3^-)}{n(NO_2) + n(HNO_3) + n(NO_3^-)}$$


Meanwhile, as an indicator of ammonia conversion efficient, ammonia transition ratio
(NTR), was calculated as the following equation (All units were $\mu g/m^3$).

$$NTR = \frac{NH_4^+/18}{NH_4^+/18 + NH_3/22.4}$$


In addition, as the fractions of ammonia, nitrate and sulfate in deliquesced aerosol, $\varepsilon$
$(NO_3^-)$, $\varepsilon(NH_4^+)$ and $\varepsilon(SO_4^{2-})$ were expressed as follows.

$$\varepsilon(NO_3^-) = \frac{n(NO_3^-)}{n(HNO_3) + n(NO_3^-)}$$


$$\varepsilon(NH_4^+) = \frac{n(NH_4^+)}{n(NH_3) + n(NH_4^+)}$$


$$\varepsilon(SO_4^{2-}) = \frac{n(SO_4^{2-})}{n(SO_2) + n(SO_4^{2-})}$$


***2.2.3 Aerosol pH***
In this work, a widely used thermodynamic model, ISORROPIA-II (Song et al., 2018; Gao et
al., 2020), was employed to establish aerosol acidity. Including the concentrations of WSIs in
$PM_{2.5}$ and gaseous pollutions (e.g., $NH_3$, HCl), the simultaneously measured temperature and
RH data were imported into its $Na^+$-$K^+$-$Ca^{2+}$-$Mg^{2+}$-$NH_4^+$-$SO_4^{2-}$-$NO_3^-$-$Cl^-$-$H_2O$ aerosol system.
According to previous study (Song et al., 2018) and our data profiles, "Forward Mode" and
"Metastable State" were selected in the model of ISORROPIA-II to calculate aerosol acidity
($H_{air}^+$, $H^+$ loading per volume air ($\mu g/m^3$)) and aerosol liquid water content (ALWC). Then the
aerosol pH was calculated by the following equation.

$$pH = -\log_{10}\frac{1000H_{air}^+}{ALWC}$$


The concentrations of $NH_3$, $NH_4^+$, $NO_3^-$ and $SO_4^{2-}$ modeled by this model significantly
correlated with their measured values with correlation coefficients of 0.971-0.999, indicating
the accuracy and acceptability of the model in this work (Fig. S1).
*2.2.4 Heterogeneous sulfate production*
Due to the necessity of precise $SO_4^{2-}$ generation, heterogeneous sulfate production ($P_{het}$) was
parameterized and calculated according to the following equation(Jacob, 2000; Zheng et al.,
2015a),
$$P_{het} = \frac{3600 sh^{-1} \times 96 gmol^{-1} \times P}{R \times T} \left( \frac{R_p}{D_g} + \frac{4}{\upsilon \gamma} \right)^{-1} S_p[SO_2(g)]$$

Where $P_{het}$ was presented in $\mu g \cdot m^{-3} \cdot h^{-1}$, $3600 sh^{-1}$ is time conversion factor, 96 g/mol is the
molar mass of $SO_4^{2-}$, P is atmospheric pressure in kPa, R is the gas constant with the value of
8.31 $Pa \cdot m^3 \cdot mol^{-1} \cdot K^{-1} \cdot$, T is the temperature with the unit of K, $R_p$ represented the radius of
aerosol particles (m), $D_g$ is the $SO_2$ molecular diffusion coefficient and $\upsilon$ is the mean molecular
speed of $SO_2$ with the typical tropospheric value of $2 \times 10^{-5} m^2 \cdot s^{-1}$ and 300 $m \cdot s^{-1}$, respectively. $\gamma$
is the uptake coefficient of $SO_2$ on aerosols, $S_p$ is the aerosol surface area per unit volume of
air ($m^2 \cdot m^{-3}$) (Jacob, 2000). $PM_{2.5}$ mass concentrations ($\mu g \cdot m^{-3}$) and mean radius (m) during
campaign were roughly calculated utilizing the following empirical formula published by Guo
et al. (2014):
$$R_p = \left( 0.254 \times C_{(PM_{2.5})} + 10.259 \right) \times 10^{-9}$$

mean density of particles $\rho$ was calculated and showed as $1.5 \times 10^6$ $g \cdot m^{-3}$ using the volume
and surface area formulas of a sphere (Guo et al., 2014). $S_p$ was estimated from the following
formula:
$$S_p = \frac{C_{(PM_{2.5})} \times 10^{-6} g \cdot \mu g^{-1}}{4/3 \cdot \pi R_p^3 \cdot \rho} \cdot 4\pi R_p^2$$

relative humidity-dependent $\gamma$ were derived according to Zheng et al. (2015a) during the
campaign in this work and showed as the following formular:
$$\gamma = \begin{cases} 2 \times 10^{-5}, & \Psi \le 50\%, \\ 2 \times 10^{-5} + \dfrac{5 \times 10^{-5} - 2 \times 10^{-5}}{100 - 50\%} \times (\Psi - 50\%), & 50\% \le \Psi \le 100\% \end{cases}$$

where $\psi$ referred to RH with the unit of %.
**3 Results and Discussion**
Based on National Ambient Air Quality Standards of China (HJ633-2012)
(https://www.mee.gov.cn/ywgz/fgbz/bz/bzwb/jcffbz/201203/t20120302_224166.shtml), air
quality index (AQI) was introduced in this work to classify pollution levels (Wang et al., 2015;

Kanchan et al., 2015; Xu et al., 2017) and discuss the characteristics of atmospheric pollutants. Briefly, daily concentrations of $PM_{2.5}$ ranged from 0-75, 75-115, 115-150, 150-250 and >250 $\mu g/m^3$ were classified as clean (C), light polluted (L), moderate polluted (M), heavy polluted (H) and serious polluted (S) periods, respectively.

### 3.1 The observed evidence for ammonia-rich atmosphere

The characteristics of atmospheric pollutants and meteorological parameters during the studied period were summarized in Supplement (S2.1). In this work, molar ratios of $NH_4^+$ vs. anions was used to identify the chemical species of ammonium salts (Zhou et al., 2018; Wang et al., 2021; Liu et al., 2017b; Shi et al., 2019). The calculated results (Supplement, S2.2) showed the predominant chemical species of ammonium gradually varied from the coexistence of ammonium sulfate $((NH_4)_2SO_4)$ and ammonium nitrate $(NH_4NO_3)$ to the coexistence of $((NH_4)_2SO_4)$, $NH_4NO_3$ and ammonium chloride $(NH_4Cl)$ with haze aggravation (Fig. S5). Further, the slope of fitted equation between excess-$NH_4^+$ and anions were still lower than 1:1 line after neutralized all the measured anions, indicating the ammonia-rich atmosphere (Fig. S5c). To meet the national demand of ultra-low emissions activities (nearly two times lower than former national standard) on gaseous pollutants, heavy usage of ammonia-containing compounds in the process of desulfurization and denitrification (Solera García et al., 2017; Tan et al., 2017) at broadly distributed thermal power plants (>300,000kWh) and the close-set coal-fired heating stations (Fig. 1) resulted ammonia fugitive provided a reasonable explanation on this ammonia-rich atmosphere. Although the retrofit of national demand of ultra-low emissions activities on gaseous pollutants (nearly two times lower than former national standard) has been completed, distributed coal-based enterprises could also emit substantial $SO_2$ and $NO_2$ and subjecting to heterogeneous reactions to further generate sulfate and nitrate and aggravated the haze events (Fig. S7a, S7b).

To show the reaction between ammonia and nitric acid and the other formation processes of nitrate in different (relative) concentrations of sulfate, the data of previous studies and different pollution levels (C, L, M, H, S) in this work were plotted in Fig. 2. When $[NH_4^+]/[SO_4^{2-}] \leqslant 1.5$, the nitrate formation associated with crustal elements rather than ammonium; while $[NH_4^+]/[SO_4^{2-}] >1.5$, the homogeneous gas-phase reactions between $NH_3$ and $HNO_3$ became the major pathway for atmospheric ammonia to form $NH_4NO_3$ (Pathak et al., 2009; Liu et al., 2019). The results illustrated that the ammonia-rich regimes were not only found in Hohhot, but also observed in Guangzhou (Huang et al., 2011), Chengdu (Huang et al., 2018), Lanzhou USA West and East, India, Ireland, Europe, Qingdao, Italy, Lin'an (Pathak et al., 2009) in recent decades (Fig. 2). It suggested that atmospheric oxidative modifications in

ammonia-rich atmosphere should be a widespread atmospheric issue with significant
contributions on SIA generation. It was worth noting that the slopes of our data were becoming
steeper, coupling with the $NO_3^-/SO_4^{2-}$ ratios change from ~4 to about 1, as the increasing
pollution levels. The high $PM_{2.5}$ nitrate concentration during Heavy and Serious stages cannot
be explained by the homogeneous gas-phase reaction involving ammonia and nitric acid, which
may be associating with the heterogeneous reaction in ALW on the surface of the preexisting
aerosols.

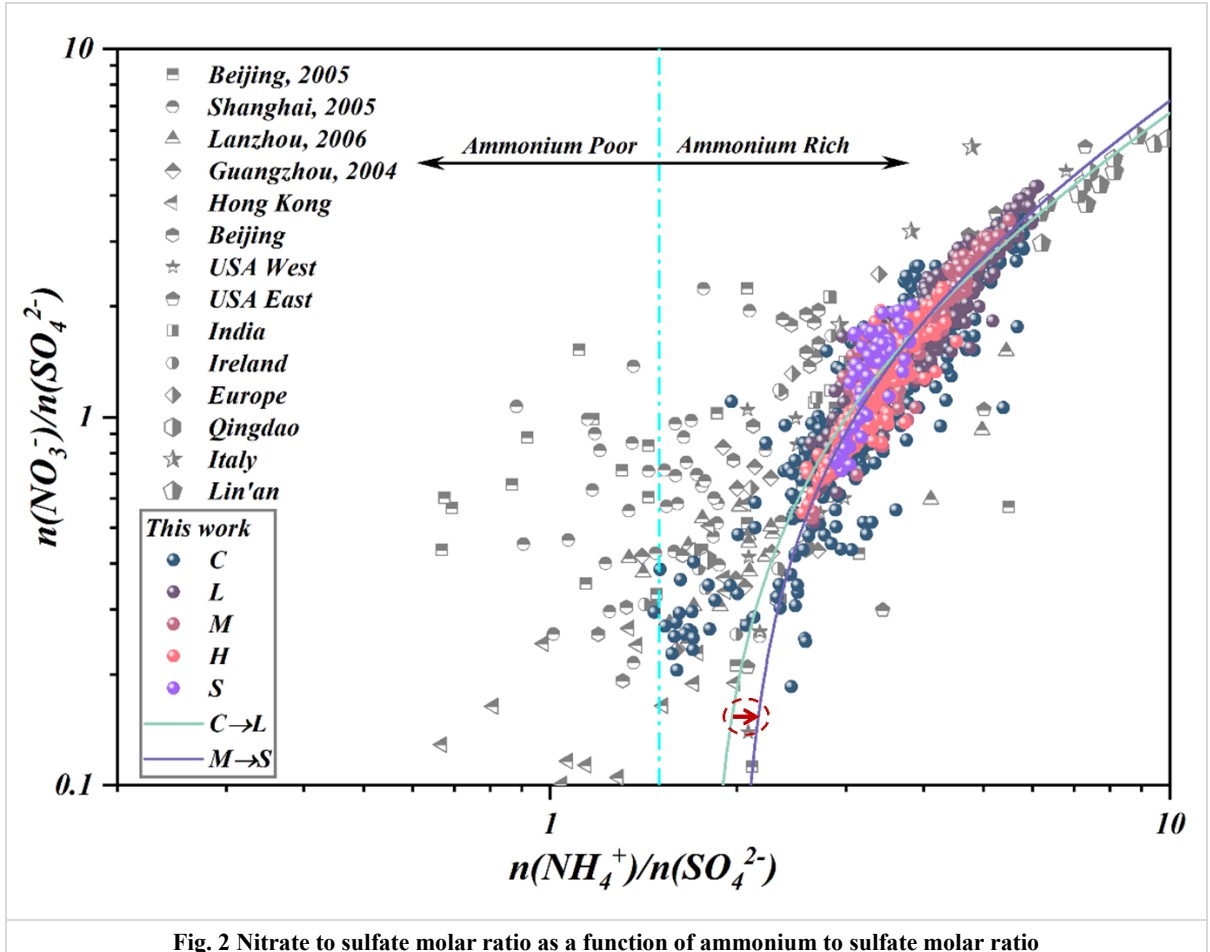

**Fig. 2 Nitrate to sulfate molar ratio as a function of ammonium to sulfate molar ratio**


## 3.2 Driving mechanism of SIAs formation

### 3.2.1 Aerosol liquid water

Our results showed that SOR, NOR and SIAs in $PM_{2.5}$ presented increasing trends with the
increasing ALWC during the five pollution levels. The variation of predominant chemical
species of ammonium (Fig. 2) indicated more SIAs will be generated on particles with the
simultaneous increase of ALWC and $PM_{2.5}$ (Fig.3b). Theoretically, the inorganic compounds
conversion was enhanced via aqueous phase chemistry on moist particles owing to the
continuous partition of gaseous pollutants (e.g., $SO_2$, $NO_2$, $N_2O_5$) in ALW, then disrupted the
equilibrium between the gaseous and condensed phases, resulting in the aggravation of haze

events (Xue et al., 2014; Wu et al., 2018; Zheng et al., 2015b; Wang et al., 2016). Considering seasonal heating characteristics, the shift of the equilibrium between gaseous and condensed phases was enhanced with the increasing atmospheric pollutants concentrations due to the coal-fired combustion events in winter. Detailly, owing to hygroscopic nature, the particles must increase their water contents via ALW along with RH (Fig. S8a) to maintain thermodynamic equilibrium and water vapor and simultaneously enhance the oxidation and dissolution of precursors in the micro-solution (ALW) of the particulates. This process elevated the inorganic mass fraction as well as particulate mass concentrations during different pollution stages (Fig. S8b) (Bertram et al., 2009; Wang et al., 2016; Zheng et al., 2015a; Cheng et al., 2016). Due to the larger affinity of $H_2SO_4$ for $NH_3$ (aq), sulfate was preferentially and fully neutralized by ammonium in the ammonia-rich atmosphere to generate non-volatile nature of $(NH_4)_2SO_4$ (Liu et al., 2017b; Zhou et al., 2018; Wang et al., 2021). Thus, SOR presented higher exponential growth with the elevated AWLC coupling with more sulfate production (Fig. 3b). Concomitantly, the preferentially generated $(NH_4)_2SO_4$ further enhanced the hygroscopicity of particulate matter, in turn, helped more ammonia partitioning into moist particulate matter and generating ammonium salts accelerating haze aggravation (Supplement, Fig. S6, Fig. S8c). Thus, most important of all, the sharp increase of inorganic compounds associating with the elevated ALWC significantly modified the specific surface area of particulates and further accelerated the hygroscopic aerosol growth, which simultaneously provided a substrate for the ensuing heterogenous reaction and accelerated the evolution of haze events. Previous work reported that particles of different modes made different contributions to ALWC with the contributions of nuclear, Aitken, accumulation and coarse modes assessed at <1%, 3%, 85% and 12%, respectively, indicating that the contribution of accumulation mode particles to ALWC dominated among all the aerosol particle modes (Tan et al., 2017). It indicated that secondary aerosol formation mainly happens on these fine particles as the surface area and volume of the fine particles are much larger than those of the coarse particles. Thus, the observed significant correlations of ALWC with the ratios ($PM_{1.0}/PM_{2.5}$ and $PM_{2.5}/PM_{10}$) in this work also indicated that the hygroscopic growth of fine particulate matter ($D_p \leqslant 2.5um$) strongly associated with ALWC (Fig. 3a). Both the previous work and our monitoring results suggested that the ratios of $PM_{1.0}/PM_{2.5}$ and $PM_{2.5}/PM_{10}$ presented the potential possibility to index the hygroscopic growth of particulate matter.

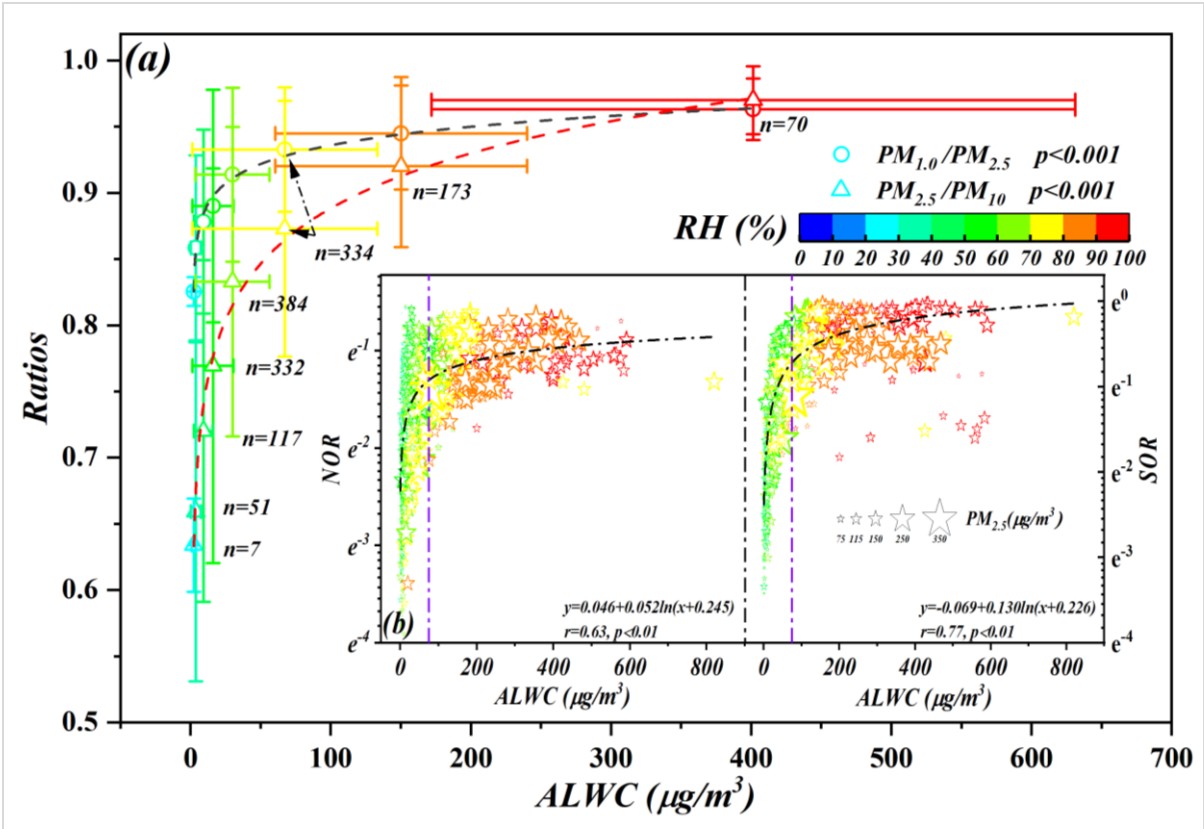

**Fig. 3 (a) Correlations between ALWC and ratios of PMs response to relative humidity evolutions, (b) Correlations between ALWC and NOR, SOR response to different pollution stages, the pentagrams were colored as a function of relative humidity**

### *3.2.2 Perturbation gases*

Due to the strict control of $SO_2$, atmospheric concentrations of $NO_2$ and $NH_3$ gradually became as the decisive reactive precursors on regional atmospheric secondary particulate matter generation. Thus, the state-of-the-art flamework proposed by Nenes et al. (2020) was carried out to exam the chemical domain classifications and the decisive precursor based on the data sets of previous studies (Nenes et al., 2020) and this work (Fig. 4). Due to the thermodynamically stable property of the preferentially generated $(NH_4)_2SO_4$, the semi-volatile $NH_4NO_3$ dominate the partitioning of $NH_3^T$ (sum of $NH_3$ and $NH_4^+$, same to $NO_3^T$) and $NO_3^T$. Although aqueous $NO_3^-$ concentrations varied with haze processes, the calculated $\varepsilon$ ($NO_3^T$) (detailed calculated method could be found in S1.2), which was an equilibrium parameter between gaseous $HNO_3$ and particle-phase $NO_3^-$ (Guo et al., 2016; Fang et al., 2017), presented consistently full loadings of nitrate on the existing particulates during the studied period (Fig. S9a, Fig. S9b). This could provide clear evidence for the initial $HNO_3$ sensitive area and continuous control of $HNO_3$ during the studied periods. However, with haze aggravation, significant elevated ALWC resulted in more precursors partitioned in micro-droplets to maintain water vapor. This process induced a positive shift of $HNO_3$ dissolution

equilibrium and leading more HNO₃ partitioned on particles driven by the Henry's law (e.g.,
$HNO_{3(g)} \leftrightarrow HNO_{3(aq)}$, $K_H$ = 2.07mol/(L·Pa)). Meanwhile, HNO₃ and HONO could also produce
through the reactions of $NO_2 + H_2O \xrightarrow{Het} HNO_3 + HONO$) (Huang et al., 2018). Accordingly,
the OH radicals generated by HONO photolysis also contributed to this oxidation processes
(Yue et al., 2020; Zhu et al., 2020). These aqueous oxidations processes were evidenced by the
observation of significantly elevated HONO and PANs during the haze aggravation
(Supplement, Fig. S7c, Fig. S7d). Accordingly, the equations of $NH_4^+ + NO_3^- + H^+ + OH^- \rightleftarrows$
$NH_4NO_3 + H_2O$ and $NH_4^+ + SO_4^{2-} + H^+ + OH^- \rightarrow (NH_4)_2SO_4 + H_2O$ were shifted to
generate more NH₄NO₃ and (NH₄)₂SO₄ (Nenes et al., 2020; Xie et al., 2020) due to the driving
force of more ammonia partitioned in elevated ALWC (NH₃+H₂O⇌NH₃·H₂O, NH₃·H₂O⇌
NH₄⁺+OH⁻). Therefore, NH₃ and NOₓ became as the decisive factors on regional atmospheric
oxidability in the ammonia-rich regime (Zhai et al., 2021; Tan et al., 2017; Liu et al., 2019; Li
et al., 2019).

Generally, both NH₃ and HNO₃ were the limiting factors governing the aerosol generations

for cities of North China due to high loadings of atmospheric ammonia, while NH₃ governed
PM formation for the southeast US (SAS) (Zhao et al., 2020). Thanks to the raw data of
Shenzhen (SZ) (Wang et al., 2022), we also calculated the ALWC and aerosol pH using
ISORROPIA-II and the scatters of SZ suggested obvious chemical transition from HNO₃-NH₃
regime to NH₃ sensitive regime due to the differently originated air masses. Although both
cities located in US, the findings in California (CNX) were quite interesting and distributed in
the insensitive region and the combined NH₃-HNO₃ sensitive region due to the moderate NH₃
levels and the complicated atmospheric conditions during the observation (Nenes et al., 2020).
In our work, the data points (541/744) in summer (pH=3.47±1.29) mostly lied in HNO₃
sensitive region, while chemical domains of perturbation gas limiting the generation of
secondary particulate matters presented obvious shifts from HNO₃ sensitive to HNO₃ and NH₃
co-sensitive regime with the haze aggravation in winter. Some data points of this work lied in
the combined NH₃-HNO₃ region in winter owing to the more acidic condition. Under the stable
pH of aerosols in winter at Hohhot (pH=4-5), the more important is that a fraction of points
will distribute in the combined NH₃-HNO₃ region when ALWC>75 μg/m³, which may be
attributed to the aqueous chemical transformation driven by Henry's law mentioned above due
to the elevating ALWC. Comparatively, the aerosols pH in summer was significantly lower
than those in winter in Hohhot. Compared to TJ and SZ, the aerosols pH of Hohhot in winter
was also significantly higher (Fig. 4) due to the acidity of atmospheric PM is largely depended
on the alkaline material in surface soils in arid and semi-arid region and the elevated
atmospheric ammonia. In terms of seasonal characteristics, the higher temperature in summer
elevates the volatility of $NH_4NO_3$ and dominates the partitioning of $NH_3^T$ in atmospheric phase
to decrease the pH of aerosols. Therefore, as can be seen from Fig. 4, the data points measured
in winter Hohhot characterized as higher pH and low ALWC than those in summer (Hohhot,
SAS, CNX, SZ). According to the framework of Nenes et al. (2020), the transition points of
Hohhot (whether winter or summer) between $NH_3$-dominated and $HNO_3$-dominated sensitivity
also occurs at a pH around 2 but at lower levels of ALWC. Theoretically, it should be associated
with the more aridity of Hohhot locating in the arid and semi-arid region of China. Our results
provided the evidence for "the additional insight" proposed by Nenes et al. (2020) that the
transition ALWC varies with season change and the aridity of sites, in response to seasonal
variability and climate change. Although this effort could provide sound explanation for
limiting gaseous pollutants on PM formation, mechanisms on their chemical domains,
especially the roles of ALW in different locations with various conditions need further study in
the future.

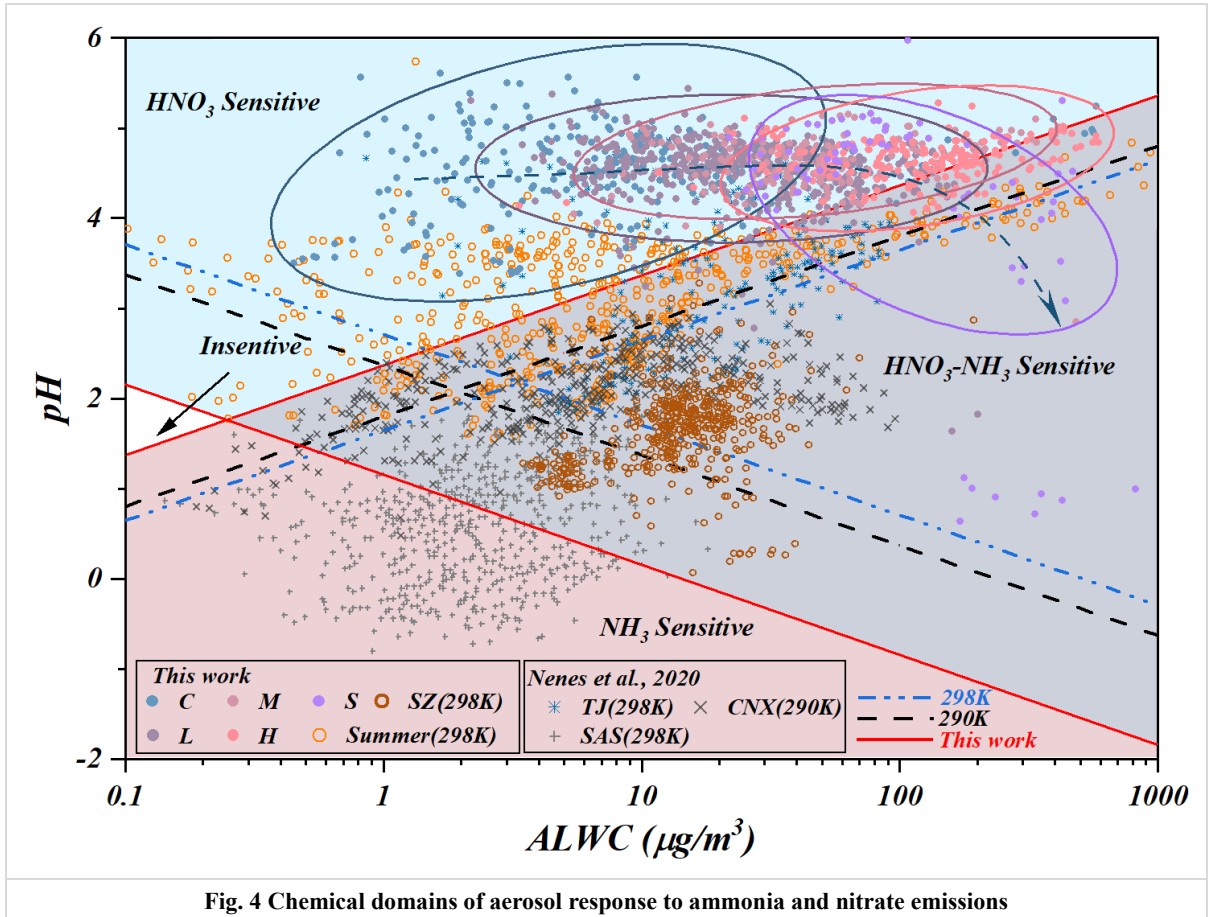

**Fig. 4 Chemical domains of aerosol response to ammonia and nitrate emissions**

*3.2.3 The shifting of SIAs formation mechanism driven by ALW*
It's worth noting that two independent correlations were found between SOR and odd oxygen
($O_x$, $O_x$=$NO_2$+$O_3$) during the aggravating processes of haze events, indicating the differential

mechanisms of atmospheric oxidability on sulfate generations at different stages (Fig. 5a). Different to inefficient homogeneous sulfate oxidation efficiency (Supplement, Fig. S10), significant correlations pairs of $NO_2$ with SOR (Fig.5b) and NOR with $SO_4^{2-}$ (Fig.5c) suggested the haze aggravation was largely related to the regional $NO_2$ levels due to the regulating effects on atmospheric oxidizability. Thus, the aqueous-phase oxidation of S(IV) by $NO_2$ (aq) was triggered and accelerated by the increasing ALWC and the following equation (Yao et al., 2020; Wang et al., 2016) (Supplement, Fig. S11a):

$$S(IV) + NO_2(aq) + H_2O \rightarrow S(VI) + H^+ + NO_2^- \tag{R1}$$

Meanwhile, sharp logarithmic increase between SOR and $NH_4^+$ were also observed from Clean to Moderate pollution stages (Supplement, Fig. S12). Due to the joint effects of ammonia-rich atmosphere and ammonia's extremely water-soluble property, sufficient hydroxide generated by ammonia dissolution forced the $NO_2$ partitioned in ALW to maintain pH through neutralization and producing sulfate via R1. Thus, the following equation (R2) was derived with considering the processes of ammonia hydrolysis, which was evidenced by Fig. S11b.

$$S(IV) + NO_2(aq) + NH_3(aq) + H_2O \rightarrow S(VI) + NH_4^+ + NO_2^- + H^+ + NO_3^- \tag{R2}$$

Generally, NOR<0.1 means insignificant nitrogen oxide oxidation, therefore the observed regime shift of nitrate and ammonia chemical behavior on sulfate generation suggested the sulfate generation was preferentially triggered by the high ammonia utilization, then accelerated by the co-effects of ammonia utilization and nitrogen oxide oxidation (Fig. 5c).

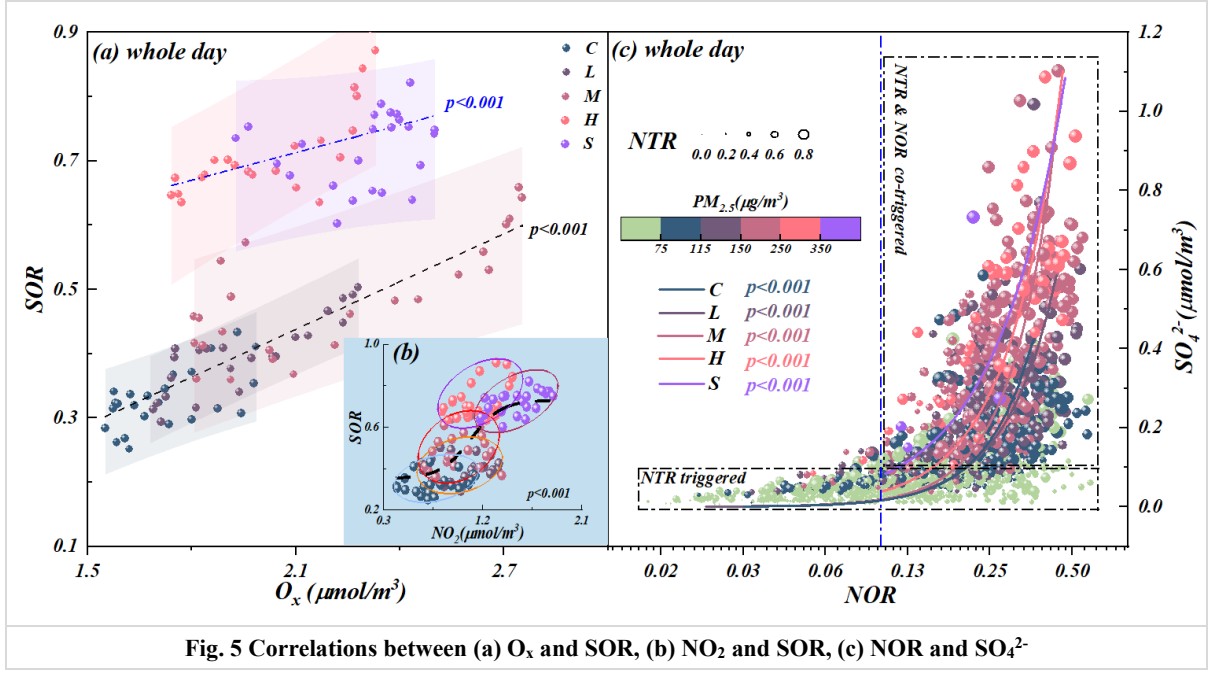

**Fig. 5 Correlations between (a) $O_x$ and SOR, (b) $NO_2$ and SOR, (c) NOR and $SO_4^{2-}$**

Accordingly, the reaction R2 was activated due to the increased ALWC forced more ammonia to partition into moist particulate matter driven by the Henry's law in the ammonia-rich atmosphere ($NH_{3(g)} \rightarrow NH_{3(aq)}$) (Supplement, Fig. S9c) (Clegg et al., 1998; Wu et al., 2018;

Xie et al., 2020). Meanwhile, our calculated aqueous generated $NO_3^-$ nicely matched theoretical nitrate aqueous generation curve (the solid blue line in Fig. S9b) proposed by Guo et al. (2017), suggesting the pathway of fast sulfate formation from oxidation of S(IV) by $NO_2$ to generate HONO (Wang et al., 2020) (Supplement, Fig. S11) via the reaction R2. As a result, the thermodynamically stable $(NH_4)_2SO_4$ would be preferentially formed to maintain its water vapor pressure and thermodynamic equilibrium, then trigged the haze formation. Thus, the mentioned effects resulted in a pronounced increase of $NH_3$ partitioning with the haze aggravation, suggesting the importance of ammonia partition on sulfate generations, namely, NTR-controlled regime with ALWC<75 $\mu g/m^3$. In summary, when ALWC<75 $\mu g/m^3$, the sulfate generation was preferentially triggered by high ammonia utilization, then accelerated by nitrogen oxide oxidation from Clean to Light pollution stages (Fig. 5c) with NOR<0.3, SOR<0.4 and NTR<0.7. In this period, the chemical composition of SIAs characterized as the moral ratio of $NO_3^-:SO_4^{2-}$=2:1 (Fig. 6).

When ALWC>75$\mu g/m^3$, the haze was aggravated from Moderate to Serious stages along with the increasing ALWC. As a result of increase in ALW, large amount of $H^+$ was dissociated during the generation of ammonium sulfate (Supplement, Fig. S13a). From Light to Moderate pollution stages, the solubility $SO_2$ driven by Henry's law was self-limiting due to the acidity effect in low ALWC (with ALWC<75 $\mu g/m^3$). Therefore, low sulfate concentrations coupled with low ALWC at the beginning of haze event (Supplement, Fig. S13a). However, due to the co-effects of elevated ALWC and hygroscopic nature of pre-generated ammonia sulfate, $H^+$ concentrations were diluted and nearly constant in-situ pH with the increase of ALWC during Heavy and Serious pollution stages (Supplement, Fig. S14) (Wang et al., 2016; Clifton et al., 1988; Huie and Neta, 1986; Lee and Schwartz, 1982). Hence, the significantly elevated ALWC provided more chance for the partition of $SO_2$, $NO_2$ and $NH_3$ in ALW from Moderate to Serious pollution stages. Theoretically, Henry's constants of $NO_2$ ($9.74\times10^{-8}$ mol·(L·Pa)$^{-1}$) is 3-4 orders of magnitude lower than those of $SO_2$ ($1.22 \times 10^{-5}$ mol·(L·Pa)$^{-1}$) and $NH_3$ ($6.12 \times 10^{-4}$ mol·(L·Pa)$^{-1}$), however, it is worth noting that the aqueous generated $NO_3^-$ from Moderate to Serious stages rapidly increased 2-5 times higher than Clean and Light stages (Supplement, Fig. S9b). Meanwhile, according to our monitoring results, the solar spectrophotometry at 380nm during Moderate to Serious stages was significantly lower than that in Clean stage (Supplement, Fig. S15), suggesting the aqueous oxidation of $NO_2$ was the predominant compared to chain photolysis (Huang et al., 2018). Accordingly, it could be deduced that aqueous-phase chemistry reaction of $SO_2$ and $NH_3$ in ALW, driven by Henry law, became the dominant mechanism for sulfate formation due to more $NO_2$ was required to take part in the

fast sulfate formation with the increase of ALWC in the ammonia-rich atmosphere by the
reaction R2. Thus, with the increasing of ALWC, high concentrations of sulfate and nitrate with
high SOR (0.5-0.9), NOR (0.3-0.5) and NTR (>0.7) induced the haze events becoming Heavy
and Serious levels (Fig. 5c). Simultaneously, the calculated heterogeneous sulfate production
rate (Jacob, 2000; Mcneill, 2015) (Supplement, Fig. S16) presented similar trends with the
impacts of ammonia on sulfate production during different pollution stages (Xue et al., 2016;
Cheng et al., 2016; Liu et al., 2020). It further stated the environmental significance of the
partitioning of $SO_2$ and $NH_3$ between gas and aqueous (ALW) phases for SIAs formation and
haze aggravation. Our results provided the evidence of significant negative correlations
between HONO and $N_2O$ (Supplement, Fig. S17) from Moderate to Serious stages and positive
correlations between HONO and SOR (Supplement, Fig.S11a), highlighting the recent reported
secondary aqueous-phase oxidation pathway of $SO_2$ by HONO from moderate pollution period
$(2N(III) + 2S(IV) \rightarrow N_2O \uparrow +2S(VI) + other\ products)$ (Wang et al., 2020). In summary,
when ALWC>75 μg/m³, aqueous-phase chemistry reaction of $SO_2$ and $NH_3$ in ALW became
the prerequisite for SIAs formation driven by Henry's law in the ammonia-rich atmosphere
during Heavy and Serious stages with high SOR (0.5-0.9), NOR (0.3-0.5), NTR (>0.7). In this
period, the chemical composition of SIAs characterized as the moral ratio of $NO_3^-:SO_4^{2-}=1:1$
(Fig. 6).

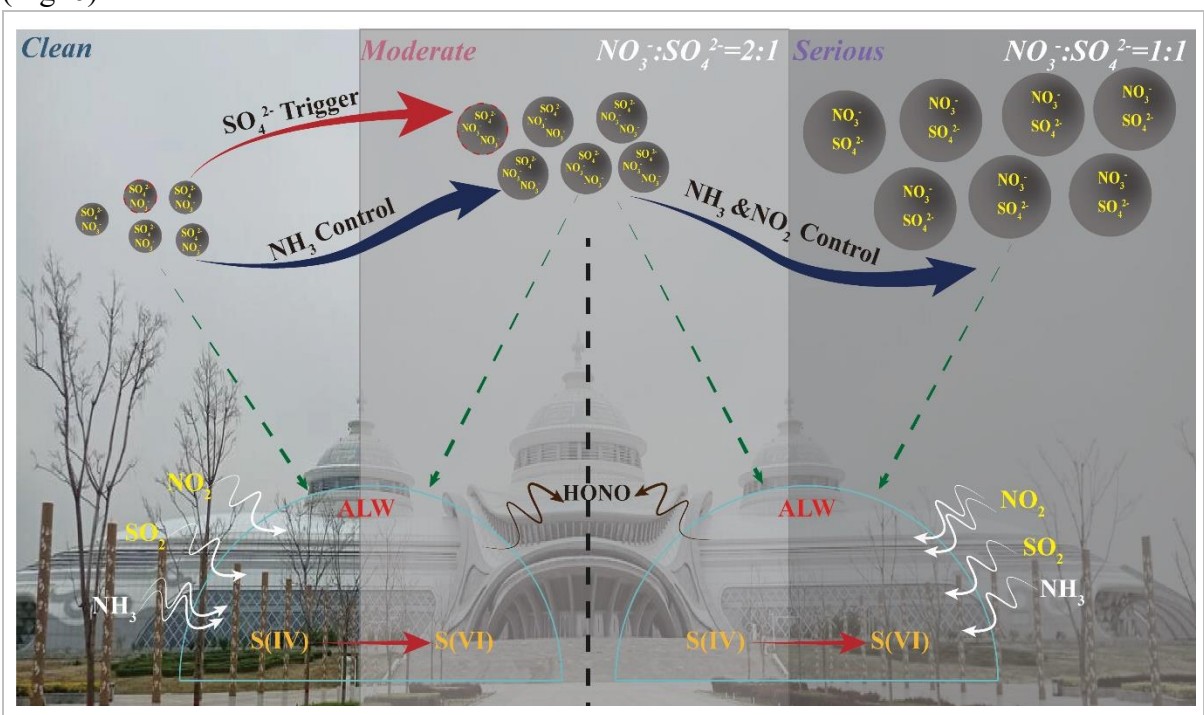

**Fig. 6 The characteristics and formation mechanism of SIAs during haze aggravation**


### 3.2.4 The positive feedback of sulfate on nitrate production

Previous works suggested that the homogeneous reaction of $NO_2$ with OH radicals during daylight and heterogeneous hydrolysis of $N_2O_5$ at night were the main routes on nitrate formation during haze episodes (He et al., 2018; Liu et al., 2020; Liu et al., 2019). Unsurprising, higher nitrate production rates ($\Delta NO_3^-$, the difference of hour concentrations and matrixing afterwards) were frequently observed in ammonia-rich conditions due to that ammonia-rich regime was more conducive on nitrate generation. However, the high level of nitrate production rates ($\Delta NO_3^-$) were found in the area characterizing as high ammonium and low sulfate levels, suggesting that highly utilizing ammonium and pre-generated sulfate promoting particle-phase nitrate generations (Fig. 7).

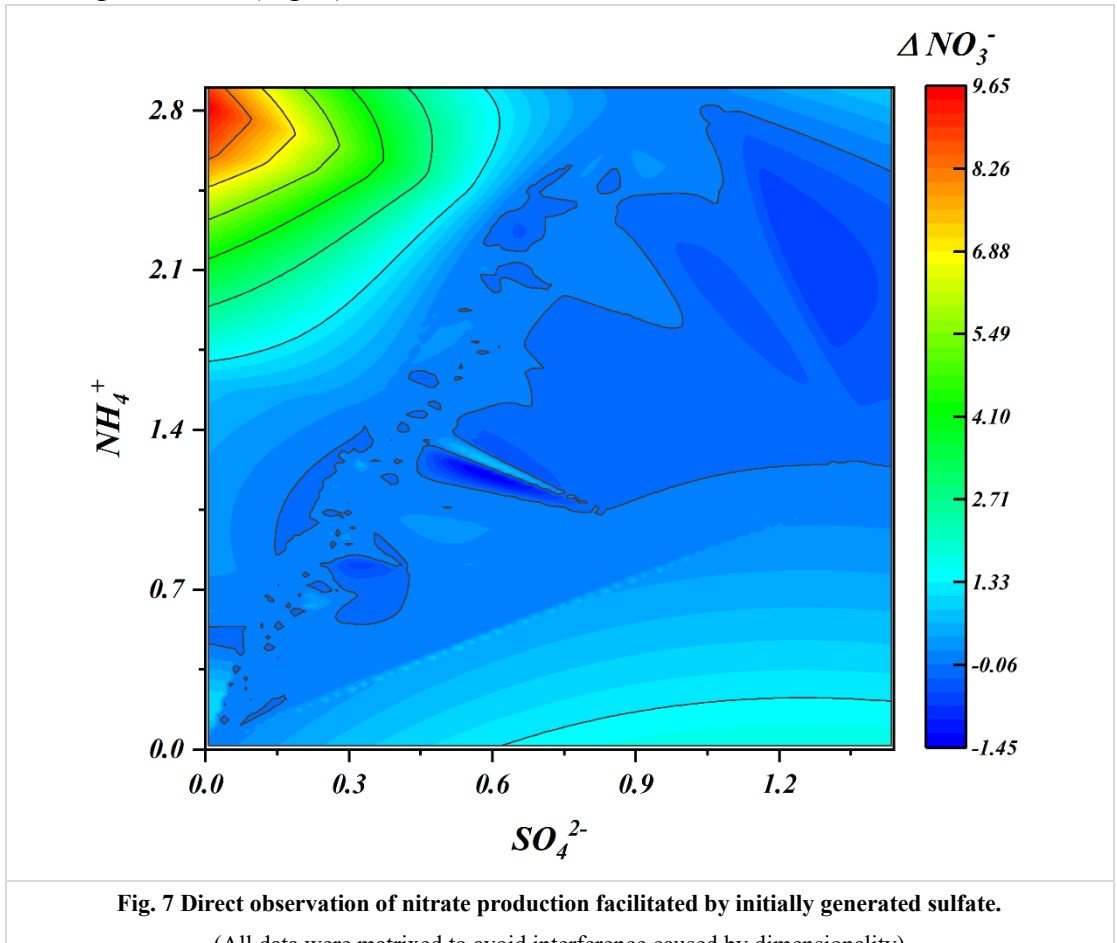

**Fig. 7 Direct observation of nitrate production facilitated by initially generated sulfate.**
(All data were matrixed to avoid interference caused by dimensionality)

Here, we proposed a hypothesis about the hydrogen ion concentration to respond the above observations. As is known to all, apart from the extremely low levels of crustal elements, ammonia is the only alkaline gas to neutralize the acidic gases in the atmosphere and generate ammonium ions (Xie et al., 2020). Thus, the concentrations of particulate sulfate and nitrate are affected by the partitioning of $NH_4^+/NH_3$. Thereby, higher values of $\Delta NO_3^-$ and $\Delta SO_4^{2-}$

always occurred in the regions with higher ammonium ions were not confused (Fig. 7, Fig. S18). According to both our results and published laboratory work (Wang et al., 2016), the acidity of the particulate matter could be significantly modified by the bulk aqueous reaction between $NO_2$ and $SO_2$, in which this reaction could be further enhanced due to in the presence of $NH_3$. As a result of the increase in RH, the partitioning of atmospheric ammonia was broken in a deep extent, which enhanced the neutralization of S(VI) by ammonia at the particle surface to generate ammonium sulfate and dissociate huge $H^+$ (Fig. S13b, red part). Simultaneously, the ALWC did not raised significantly (Fig. S14b) at the beginning of haze event with relative low sulfate concentrations. Thus, hydrogen ions generated from sulfate dissociation absorb ammonia more effectively from the ammonia-rich atmosphere at low relative humidity during the early pollution stages, which significantly promotes the net nitrate production. However, due to the co-effects of elevated RH and hygroscopic nature of pre-generated ammonia sulfate, $H^+$ concentrations were diluted and shown as nearly constant in-situ pH (Fig. S14a). According to previous works, the reaction between firstly generated sulfate and bisulfate with ammonia were treated as the determination reaction on particle acidity (Weber et al., 2016; Liu et al., 2017a). This reaction is self-limiting due to the acidity effect, namely that it increases the acidity of aqueous phase and in turn reduces the efficiency of Henry's constant for $SO_2$ solubility and reaction rate and reduced the $H^+$ formation rates from moderate periods, compared with clean periods (Fig. S13b, blue) (Wang et al., 2016; Clifton et al., 1988; Huie and Neta, 1986; Lee and Schwartz, 1982). Due to the co-effects of RH increase and hygroscopic of sulfate, the ALWC was significantly elevated with the worsen of haze. Although more $H^+$ was generated in this process, no significant decrease in pH was found with the haze aggravation due to the dilution effect of ALWC on $H^+$. Previous works suggested that in the case of ALWC increase, nitrate production is controlled by elevated $H^+$ associating with the increase of sulfate, namely, $NO_3^-$ presented elevating trend with the increases of $H^+$ concentration (Xie et al., 2020). Thus, although $H^+$ from the dissociation of sulfuric acid and full-loaded particle nitrate in conjunction with the haze aggravation generate particle $HNO_3$ (Fig. S19a) could forcing more ammonia partitioned on the particles to generate ammonium nitrate (Fig. S19b), net nitrate production ($\Delta NO_3^-$) was nearly consistent.

## 4 Conclusions

The formation of SIAs, especially sulfates and nitrates, was inherently associated with ALWC during the haze aggravation, in which the roles of ALWC should be more significant in ammonia-rich atmosphere. The novelty of our work is to find the shifting of secondary inorganic aerosols formation mechanism during haze aggravation and explain the different

roles of ALWC in a broader scale ($\sim$500 ug/m$^3$) in ammonia-rich atmosphere based on the in-situ high-resolution on-line monitoring data sets. The results showed that chemical domains of perturbation gas limiting the generation of secondary particulate matters presented obvious shifts from HNO$_3$ sensitive to HNO$_3$ and NH$_3$ co-sensitive regime with the haze aggravation, indicating the powerful driving effects of ammonia in ammonia-rich atmosphere. When ALWC<75 ug/m$^3$, the sulfate generation was preferentially triggered by the high ammonia utilization, then accelerated by nitrogen oxide oxidation from Clean to Moderate pollution stages, characterizing as NOR<0.3, SOR<0.4, NTR<0.7 and the moral ratio of NO$_3^-$:SO$_4^{2-}$=2:1. While ALWC>75 ug/m$^3$, aqueous-phase chemistry reaction of SO$_2$ and NH$_3$ in ALW became the prerequisite for SIAs formation driven by Henry's law in the ammonia-rich atmosphere during Heavy and Serious stages, characterizing as high SOR (0.5-0.9), NOR (0.3-0.5), NTR (>0.7) and the moral ratio of NO$_3^-$:SO$_4^{2-}$=1:1. A positive feedback of sulfate on nitrate production was also observed in this work. Our work provides a potential explanation for the interactive mechanism and feedback between nitric aqueous chemistry and sulfate formation in ammonia-rich atmosphere based on high-resolution field observation. It implies the target controlling of haze should not simply focus on SO$_2$ and NO$_2$, more attention should be paid on gaseous precursors (e.g., SO$_2$, NO$_2$, NH$_3$) and aerosol chemical constitution during different haze stages.

*Data availability.* All data of this study are available from the corresponding author upon reasonable request (lcw2008@imu.edu.cn).

*Supplement.* The Supplement related to this article is available online at

*Author Contributions.* FX: Data curation, Formal analysis, Software, Writing-original draft. YS: Investigation, Formal analysis. YLT: Methodology, Software. YSH: Investigation, Formal analysis. XJZ: Investigation, Formal analysis, Software. PW: Methodology, Investigation. RHY: Software, Writing-review & editing. WW: Investigation, Validation, Writing-review & editing. JH: Investigation, Methodology. JYX: Investigation, Validation, Supervision, Writing-review & editing. CWL: Initiating and leading this research, Supervision, Writing-review & editing.

*Competing interest.* The authors declared that they have no conflict of interest.

*Acknowledgments.* This work is supported by Science and Technology Major Project on Air Pollution Prevention and Prediction in Hohhot-Baotou-Ordos Cities Group of Inner Mongolia (No. 2020ZD0013), National Natural Science Foundation of China (No. 42167028, 41763014) and Science Fund for Distinguished Young Scholars of Inner Mongolia (2019JQ05).

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
