# Peer review of "The shifting of secondary inorganic aerosols formation mechanism during haze aggravation: The decisive role of aerosol liquid water"

_Atmospheric Chemistry and Physics, 2022_

## Author Comment (AC1)

**Response to the reviewers**

Dear reviewers and editors,

Our manuscript entitled "The shifting of secondary inorganic aerosols formation mechanism during haze aggravation: The decisive role of aerosol liquid water (acp-2022-590)" has been revised according to the comments raised by the reviewers. We are very thankful the comments from the reviewers and editors which largely improved the quality of our manuscript. All changes have been highlighted using light blue in the revision. The detailed and point-by-point response to the reviewer comments were detailed below.

***Reviewer 1[#]:***

*General comments*

*Aerosol liquid water is an important constituent in $PM_{2.5}$, which has a significant impact on the secondary aerosols. This manuscript reported a long-term observation of $PM_{2.5}$ and estimated ALWC and pH. And a detailed analysis on their relationships in different pollution conditions was performed. It is fully within the journal scope, but the authors should resolve the following questions before it can be published on ACP.*

*Major comments:*

*1. The English should be improved. Some of the sentences are too long to follow. Also the format of units should be revised. L52 in the abstract "$\mu g/m^3$". L105, the format of $\mu g/m^3$... Please unify the units in the manuscript.*

**Response:** Thanks a lot. We have carefully checked and unified the units format throughout the full text. Meanwhile, we have checked the English expression of the manuscript and rewritten or restructured some long sentences according to your suggestions.

*2. Throughout the whole text, you are saying the aqueous phase reactions and secondary formation in the local area. I am doubting whether there is long-range transport or primary emission resulting in the accumulation of secondary inorganic aerosols? And also the boundary layer change can also affect the concentration. Did you consider these factors other than secondary formation?*

**Response:** Thanks for your comments. We agree with you that the long-range transport, primary emission and the boundary layer change are also the important factors on the accumulation of secondary inorganic aerosols. According to our monitoring results, the effects of the mentioned three factors on the accumulation of secondary inorganic aerosols may not be the dominant, and the reasons are as follows.

Firstly, large scale regional haze pollution events actually occurred simultaneously at Tumocheon Plain (including Baotou city) during our studied periods. Whereas, according to the environmental monitoring results, the average wind speed was only 1-1.15 m/s during Moderate, Heavy and Serious pollution stages (Table S1 in supplement), which denied the possibility of long-range transport as the dominant factor on the accumulation of secondary inorganic aerosols.

Secondly, from Heavy to Serious pollution stage, the concentration of $NO_2$ elevated from 47 to 57 $\mu g/m^3$, coupling with the increase of $SO_2$ from 15 to 33 $\mu g/m^3$, which corresponded to the elevating levels of $PM_{2.5}$ from 150-192 $\mu g/m^3$. The most important was that the observed levels of $PM_1$ were elevated nearly two times (90 to 170 $\mu g/m^3$) from Heavy to Serious pollution stage. Although the concentration of $PM_{10}$ did not increase significantly (from 177 to 200 $\mu g/m^3$), the grain size characteristics presented significant changes, in which the ratio of $PM_1/PM_{10}$ increased from 50 to 85% during Heavy to Serious pollution stage. These results directly showed more fine particles were generated in this period indicating that the contributions of the aqueous phase reactions and secondary formation were higher than primary emission in the local area in this work. Meanwhile, concentration weighted trajectory (CWT) and 24-h backward trajectory clustering analysis were performed during the observation period (Fig. i). The 24-h backward trajectory clustering results suggested that the air mass during the studied period were local originated (37.37%, trajectory 1), northern Ordos (32.86%, trajectory 2) and southern Mongolia (29.77%, trajectory 3). Accompanied with backward air mass trajectory, the CWT results suggested that the air mass originated from southern Mongolia (trajectory 3) was clean air. As a result, trajectory 2 (northern Ordos) might be the only path for the regional transmission of $PM_{2.5}$, if possible. According to the definition of Potential Source Contribution Factor (PSCF) analysis, $m_{ij}$ is retention time of the trajectories with threshold pollutant in cell ij. According to Fig. 1b, the air mass around the sampling site presented the longest retention time with characteristics in accordance with trajectory 1. Thus, it suggested that the haze events were much likely originated from local scale, with less

contributions of long-distance transport (trajectory 2).

Thirdly, in terms of the boundary layer change, the calculated boundary layer heights were almost same during Heavy and Serious stages (340 and 337 m, respectively) in our studied periods. As mentioned above, the elevation of $PM_{2.5}$ levels (150-192 μg/m$^3$), especially $PM_1$ (90 to 170 μg/m$^3$), should not be resulted from the boundary layer change.

[Figure]

Fig. i (a) 24-h backward trajectories, concentration weighted trajectory and (b) $m_{ij}$ results during studied periods

*3. Section 2.2 I doubt in the estimation of Rp and Sp. You cited the previous studies which conducted in different locations? Is it representative for your site?*

**Response:** Thanks for your valuable comment. In fact, particle morphology is very complex and varies significantly. Thus, in order to carry out the theoretical research, the scientific community uniformly regards the particle as ideal spheres. Accordingly, we employed the reported empirical formula published by Guo et al. (2014) to calculate aerosol mean radius and surface area, which was also employed in several works, such as North China Plain (He et al., 2018; Shao et al., 2019; Zheng et al., 2015). We are very grateful for your comments which guide us to do more work on the characterization of mean radius and surface area and the modification of the formula parameters for different cities or regions.

*4. L189-191. I don't know how to get the conclusion about the homogenous gas-phase reaction from Fig. 2. Please explain it.*

**Response:** Thanks a lot. These sentences had been revised in the revision

according to your comments. Previous studies suggested that the nitrate-to-sulfate molar ratio ($[NO_3^-]/[SO_4^{2-}]$), as a function of the ammonium-to-sulfate molar ratio ($[NH_4^+]/[SO_4^{2-}]$), could be used to indicate the pathways of nitrate formation in different atmospheric compositions (Pathak et al., 2009; Pathak et al., 2004). When $[NH_4^+]/[SO_4^{2-}] \leqslant 1.5$, the nitrate formation associated with crustal elements rather than ammonium due to the preferential formation of ammonium sulfate and ammonia bisulfate with a stoichiometric ratio of about 1.5. In ammonium poor samples, pseudo-first-order kinetics results showed the rate of neutralization of acidity were extremely higher than the rate of ammonium nitrate formation because neutralization of acidity was the major pathway for atmospheric ammonia. However, when $[NH_4^+]/[SO_4^{2-}] > 1.5$, ammonium nitrate formation rate sharply raised to a higher level, while the acidity neutralization reaction kept a relative equilibrium (Fig. ii). In summary, neutralization of acidity constrains the uptake of atmospheric ammonia in ammonium poor conditions, while ammonia reacts with nitric acid to form ammonium nitrate in ammonium rich atmosphere.

[Figure]

Fig. ii Rates of ammonium nitrate formation and in situ acidity neutralization and the maximum rate of acidity neutralization function of ammonium to sulfate ratio molar ratio (Huang et al., 2011)

5. *L194-195. You said the ammonia has an important role in the atmospheric oxidative modification. Please listed the supporting information here.*

**Response:** Thanks for your suggestions. Previous studies suggested that ammonia may enhance atmospheric HONO formation, a representative oxidizing gas, by

promoting the disproportional hydrolysis of N₂O₅. Meanwhile this reaction could be enhanced because NH₃ could markedly reduce the free-energy barrier of N₂O₅ hydrolysis into HONO and HNO₃. Additionally, ammonia is able to partition into the particle phase via direct dissolution owing to its high aqueous solubility and by neutralize the acids materials due to its stronger base nature. Accordingly, both previous works and our results highlighted the important effects of ammonia on the oxidation modification of secondary aerosol in ammonia-rich atmosphere (Malloy et al., 2009; Atkinson et al., 1997; Wang et al., 2015; Jiang and Xia, 2017; Li et al., 2018).

*6. L209-213. Is there any possibility that the homogenously formed NH₄NO₃, (NH₄)₂SO₄ partitioned into the aqueous phase?*

**Response:** Thank you very much. A large number of field observations and laboratory work reported that the formation of (NH₄)₂SO₄ is mainly attributed to heterogeneous reactions due to the non-volatile nature of H₂SO₄ and (NH₄)₂SO₄ and presented as follows (Hung and Hoffmann, 2015),

$$SO_{2(g)} + H_2O \overset{K_H}{\Leftrightarrow} SO_2 \cdot H_2O_{(aq)}, K_H = 1.3 mol\ kg^{-1}\ bar^{-1}$$
$$SO_2 \cdot H_2O \overset{K_{a1}}{\Leftrightarrow} HSO_3^- + H^+, K_{a1} = k_1/k_{-1} = 10^{-1.8}M$$
$$HSO_3^- \overset{K_{a2}}{\Leftrightarrow} SO_3^{2-} + H^+, K_{a2} = k_2/k_{-2} = 10^{-7.2}M$$
$$SO_3^{\bullet-} + O_2 \rightarrow SO_5^{\bullet-}, k = 1.5 \times 10^9 M^{-1}s^{-1}$$
$$SO_5^{\bullet-} + SO_3^{2-} \rightarrow SO_4^{2-} + SO_4^{\bullet-}, k = 1.3 \times 10^7 M^{-1}s^{-1}$$
$$SO_4^{\bullet-} + SO_3^{2-} \rightarrow SO_4^{2-} + SO_3^{\bullet-}, k = 2.0 \times 10^9 M^{-1}s^{-1}$$
$$SO_4^{2-} + NH_4^+ \rightarrow (NH_4)_2SO_4$$

Thus, it would like to suggest that heterogeneous reaction was the dominant mechanism for (NH₄)₂SO₄ generation. Furthermore, the nitrate formation driven by the homogeneous reaction below depends heavily on temperature and relative humidity (Mozurkewich, 1993).

$$HNO_{3(g)} + NH_{3(g)} \leftrightarrow NH_4NO_{3(aq)}$$

In this equation, the theoretical dissociation constant $K_p^*$ (ppb²) for NH₄NO₃ can be calculated by the following empirical equation (Tao et al., 2016; Wang et al., 2019; Seinfeld et al., 1997).

$$K_p^* = (P_1 - P_2(1 - a_w) + P_3(1 - a_w)^2) \times (1 - a_w)^{1.75} K_p$$

Where $P_1$ to $P_3$ are the empirical constants and $a_w$ is the water activity. According to the observed average HNO₃ and NH₃ concentrations, the average values of $K_p^*$ were

calculated as 0.61, 0.81, 1.39, 1.56 and 2.80 $ppb^2$ during Clean, Light, Moderate, Heavy and Serious stages, respectively. While the values of $K_p^*$ (0.025, 0.013, 0.011, 0.003 and 0.006 $ppb^2$ for Clean, Light, Moderate, Heavy and Serious stages, respectively) were significantly lower during our observation periods, indicating the homogeneous reaction was not the dominant mechanism on nitrate formation in our studied periods.

*7. L250-254. I'm confused by this sentence. Are you sure the oxidation was taken place in the aqueous phase initiated by the protons or it is just a thermal equilibrium from the gas phase to the aqueous phase?*

**Response:** Thanks for your comments. We are so sorry that this sentence confused you so much. This sentence had been revised according to your comments.

*8. L265-270. Did you try to find any data from China? Since your data was from a Chinese site, I think it would be better to compare the other sites from China. Additionally, it is better to cite some foreign sites, like USA.*

**Response:** Thank you very much for your valuable suggestions. Accordingly, the data from Chinese cities of Tianjin (TJ) and Shenzhen (SZ) and foreign sites of southeast USA and California were cited in the revision to discuss the difference. The details had been updated in Fig. 4 in the revision.

*9. Why did you choose the wavelength at 380 nm? Is it efficient for photochemistry?*

***Response:*** Thanks a lot. The bond energy of $NO_2$ is 300.5kJ/mol and it is observed that $NO_2$ has a continuous spectrum at the range of 290-380 nm in the troposphere ($NO_2 + h\nu(290{\sim}380nm) \rightarrow NO + O \cdot$). Meanwhile, the photolysis rate of $NO_2$ could be calculated by the following equation (Tang et al., 2018),

$$r_{NO_2} = \sum_{\lambda=290}^{\lambda=380} \sigma(\lambda)\Phi(\lambda)J(\lambda)d\lambda[NO_2], \ I(\lambda) = \sigma(\lambda)J(\lambda)[NO_2].$$

Accordingly, $r_{NO_2} = \sum_{\lambda=290}^{\lambda=380} \Phi(\lambda)I(\lambda)d\lambda$. Besides, the corresponding photolysis rate parameters showed an increasing trend from 290 to 380 nm and peaked at 380 nm. Thus, considering the limitation of solar spectrophotometer, we selected 380 nm as a parameter to indicate $NO_2$ photolysis.

*10. Section 3.2.4. A major concern here is that how can I distinguish the secondary formation and transport of $NO_3^-$ just from $\Delta NO_3^-$. It seems that you use $\Delta NO_3^-$ to represent the secondary formation of $NO_3^-$. I can't agree with you about this.*

**Response:** Thank you for providing this valuable suggestion. Just as you mentioned in your comment 2, the long-range transport, primary emission and the boundary layer change are also the important factors on the accumulation of SIAs. The SIAs composition and levels mainly depended on secondary formation, which has been replied in details in comment 2. Accordingly, the results obtained by minusing method were considered to represent the secondary formation of $NO_3^-$ were reliable. Similar method was applied in other studied areas on mechanism discussion, such as aerosol acidity and sulfate formation mechanisms during sand storm periods (Jia et al., 2020; Sharma et al., 2022; Wang et al., 2014).

*Minor comments:*
*Main text:*

*1. Some of the abbreviations in the main text are confused to me. Please list their full names when they appeared for the first time. And the figures in the manuscript and SI should be reorganized since some of them are too small to see them clearly, like Fig. 5b and Figs. S3, S8 and S9.*

**Response:** Thank you very much. According to your suggestions, we have carefully checked the abbreviations and supplied their full names when they appeared for the first time. Further, some figures were reorganized to make them easier to see.

*2. Did you measure the transition metal ions during your campaign? If yes, it is better to show the heavy metal data than just cite the literatures. Or I don't think it is necessary to talk about heavy metals.*

**Response:** Thanks a lot. We agree with you and this sentence had been deleted.

*3. Please show the exact pH value in summer.*

**Response:** The exact pH values in summer had been provided accordingly.

*4. This figure is quite difficult for me to follow. It is better to simplify it and become easier to understand.*

**Response:** Thanks. This figure had been redrawn and simplified according to your suggestions which made make it easier to understand for readers.

*5. The Z axis is repetitive with the color bar.*

**Response:** Thank you very much. This figure had been changed from 3D to 2D plot.

*Supplementary:*

*1. Most of the abbreviations are defined in SI. I think you should move them into the main text.*

**Response:** Thank you very much. The abbreviations had been moved to the main text in the revision.

*2. Fig. S8b didn't show the $NO_3$ concentration. It is SIA.*

**Response:** Thanks. In the original version, $NO_3^-$ concentration were placed at Fig. S9b rather than Fig. S8b. Accordingly, we have carefully checked all figures in the manuscript and supplement materials.

*3. As a main part of your manuscript, Table S1 should contain ALWC in different stages.*

**Response:** Many thanks. The data of ALWC had been supplied in Table S1 according to your suggestion.

***Comments by Reviewer 2[#]:***

*The aerosol liquid water plays a profound role in secondary inorganic aerosols. reported. Fully understanding ALW and its roles are fundamentally important in atmospheric physicochemical processes, especially the liquid chemical transformation of $SO_2$ and $NO_x$. The manuscript gave measurement results of the secondary inorganic aerosol properties under different aerosol liquid water content. The results are interesting and the manuscript can be published after denoting the following comments.*

*Major Comments:*

*1. This manuscript is submitted as a full research paper to the ACP. The data acquisition and analysis method should be placed in the main manuscript but not in the supplement. At the same time, some calculating methods and definitions were not described in the manuscript.*

**Response:** Thanks a lot. The data acquisition, analysis method, some calculating methods and definitions had been provided in the main manuscript according to your

suggestions, which make the revision easier to understand for all readers.

*2. The subfigures in figure 3, figure 5, and some others in the supplementary materials should be in parallel with the main one. I recommend these figures be reorganized.*

**Response:** Thanks for your valuable suggestions. In these figures, the subfigures are used to help better understand the main ones. Thus, some figures are still kept in their original version. Thank you again.

*3. Section 3.1 noted that 2019). The calculated results (Supplement, S2.2) showed the "predominant chemical species of ammonium gradually varied from the coexistence of ammonium sulfate $((NH_4)_2SO_4)$ and ammonium nitrate $(NH_4NO_3)$ to the coexistence of $((NH_4)_2SO_4)$, $NH_4NO_3$ and ammonium chloride $(NH_4Cl)$ with haze aggravation". I'm not convinced by this conclusion. The author should provide the SIA ratios (including $Cl^-$) under different pollution levels. As shown in Fig.S3, the $Cl^-$ also exists during the clean periods. The variation of pH under different pollution levels should also be given.*

**Response:** Thanks for your comments which help us to improve the understanding of the chemical species of SIAs in $PM_{2.5}$. Generally, molar ratios of $NH_4^+$ vs. anions was widely used to identify the chemical species of ammonium salts (Zhou et al., 2018; Wang et al., 2021; Liu et al., 2017; Shi et al., 2019). In this work, this method was also employed to deduced the dominant species of ammonium which was detailed in Supplement (S2.2, Fig. S5a and Fig. S5b). According to your comments, more details were provided in section S2.2 and the SIA ratios (including $Cl^-$) under different pollution levels were also provided in Tables S2. Additionally, pH under different pollution levels were given in Table S1. Thanks for your comments again.

*4. The author concluded that the process in lines 216 to 218 from the fact that HONO and PANs elevated with the haze aggravation. However, I think the variation of the ratio of $HONO/PM_{2.5}$ and $PANs/PM_{2.5}$ under different pollution levels should be the proxy of their conclusion. The increase of HONO and PANs with the pollution levels can be attributed to the accumulation of pollution precursors and this pollution cannot be diluted.*

**Response:** Thank you very much. We agree with your valuable suggestions that the ratios of $HONO/PM_{2.5}$ and $PANs/PM_{2.5}$ should be the better proxies to characterize

the different pollution levels. In fact, during the data process, we had tried our best to calculate the ratios of HONO/PM$_{2.5}$ and PANs/PM$_{2.5}$ (Fig. iii). The results were plotted below. As can be seen from the figure, the ratios of HONO/PM$_{2.5}$ and PANs/PM$_{2.5}$ do not work well due to the trends of the ratios depend on both numerator and denominator. Accordingly, the trends of HONO/PM$_{2.5}$ and PANs/PM$_{2.5}$ depend on not only the variations of the concentrations of HONO and PANs, but also PM$_{2.5}$. Based on the monitoring results during the studied periods, the PM$_{2.5}$ concentrations varied from 39.11±31.1 to 192.14±162.9 μg/m$^3$ (Table S1), almost 5 times in serious stage than that in clean stage, which the rangeability is obviously higher than those of HONO and PANs. Thus, the mass concentration of HONO and PANs are still used in this work.

[Figure]

Fig. iii The trends of HONO/PM$_{2.5}$ and PANs/PM$_{2.5}$ under different pollution levels

*Minor Comments:*

*1. Line 24, the NOR, SOR, and NTR were not defined in the abstract.*

**Response:** Thanks a lot. The abstract had been revised according to your comment.

*2. Line 30-32, I got what the author means, but I think it should be noted that the NH$_3$ should also be concerned during the severe haze stage.*

**Response:** Thank you very much. According to your suggestions, this sentence had been revised to address the importance of chemical shift on ammonia.

*3. Line 58, it is not clear why "Therefore, it is urgent to fully understand the chemical regimes and behavior of reactive gases during different pollution stages and propose reasonable strategies." Some explanations should be given.*

**Response:** Thanks a lot. This sentence had been revised according to your comments, which make it more clear for readers.

*4. Line 76, the heating season at this place is not appropriate, as it is not mentioned before in the introduction.*

**Response:** Thanks for your suggestions. This sentence had been revised.

*5. The S1.1.1, S1.1.2, and S1.1.4 should be placed in the main manuscript.*

**Response:** Thank you very much. Section *S1.1.1, S1.1.2, and S1.1.4* had been moved to main manuscript according to your comments.

*6. Line 129, the definition of different classified pollution levels should be placed in the main manuscript as the following parts referred to the definition many times.*

**Response:** Thank you very much. The definition of different classified pollution levels had been placed in the main manuscript according to your suggestions.

*7. Line 154, how was the conclusion get from figure 2?*

**Response:** Thank you very much. The same comment had been raised by Reviewer 1[#] (comment 4). Pls. see our response and explanation above.

*8. Line 196-197, please explain why the ratios ($PM_{1.0}/PM_{2.5}$ and $PM_{2.5}/PM_{10}$) can be used as the proxy of the hygroscopic growth of particulate matter.*

**Response:** Thanks for your comments. Previous work suggested that particles of different modes made different contributions to ALWC with the contributions of nuclear, Aitken, accumulation and coarse modes assessed at <1%, 3%, 85% and 12%, respectively, indicating that the contribution of accumulation mode particles to ALWC dominated among all the aerosol particle modes (Tan et al., 2017). Furthermore, significant correlations of ALWC with the ratios ($PM_{1.0}/PM_{2.5}$ and $PM_{2.5}/PM_{10}$) were observed in this work (Fig. 3(a)). Therefore, the hygroscopic growth of fine particulate matter ($D_P \leqslant 2.5um$) strongly associated with ALWC. Accordingly, both the previous work and our monitoring results suggest that the ratios of $PM_{1.0}/PM_{2.5}$ and $PM_{2.5}/PM_{10}$

can be used as the proxy of the hygroscopic growth of particulate matter.

This sentence had been revised in the revision according to your comments. Thank you again.

*9. Line 208, the method of calculating the ε (NO$_3^T$) should be given.*

**Response:** Thanks for your suggestions. The calculating method of ε (NO$_3^T$) were provided in the Supplement accordingly.

*10.Line 235, the number of points should be given.*

**Response:** Thanks a lot. The number of the points had been given in the revision.

*11. Figure 7, the 3D plot is not necessary as the fill color can give the results.*

**Response:** Yes, we agree with you. This figure had been changed from 3D to 2D plot.

*Some minor comments on the supplements:*

*1. I'm not smart enough to get the meaning of the caption in Figure S9(a).*

**Response:** We are very sorry that this figure confused you so much. To be honest, we're also not entirely satisfied with the presentation of this graph. Accordingly, it had been redrawn in the revised Supplement. In details, the height of the columns (C, L, M, H and S) means the number of hourly-monitoring values during each pollution stage; while the width of the columns (C, L, M, H and S) stands for the loading capacities.

*2. S2.2, the first two lines were not clear.*

**Response:** Thank you very much. This sentence had been revised according to your comments which made it more clear for readers.

***Comments by Reviewer 3#:***

*The manuscript offered a fully discussion on atmospheric interest, the formation mechanisms of secondary inorganic aerosols during a month-lasting haze periods by occupying the scientific stoichiometry methods. The manuscript is nicely constructed and the result presented in this work is relevant because, unlike the previous works, this work not only highlights the importance of both reaction medium and its corresponding gaseous precursors on secondary inorganic aerosol generations, but also identifies the*

*chemical behavior transformation of gaseous precursors during haze aggravation. The findings of this manuscript could provide a new insight on secondary inorganic formation mechanisms. Nevertheless, some details must be addressed before it is accepted for publication at ACP.*

*1. Many μg/m³ were mistakenly written as ug/m³, pls. carefully checked.*

**Response:** Thanks a lot. We had carefully checked and revised the incorrect unit presentation in the manuscript accordingly.

*2. There were many abbreviations in this manuscript, pls. give the full-length of the words at their first appearance.*

**Response:** Thank you. All abbreviations had been given the full-length of the words at their first appearance according to your comments.

*3. Some important effects of haze pollution should be mentioned in the introduction part as the motivation and urgency, such as climate and health.*

**Response:** Thanks for your suggestions. The relevant contents had been supplied in the part of introduction.

*4. Line 235, the data points in summer mostly lied in $HNO_3$ sensitive region. How is this defined? I can hardly find the sensitive domain boundaries during summer time.*

**Response:** Thanks for your comments. Generally, the boundaries of sensitive regimes were calculated by the temperature. The average temperature was 21.91±4.28°C (295 K) during summer in our studied area, which was near to 298 K (Fig. iiii). In order to simplify the figure, only the regime boundaries of 298K were plotted in Fig. 4 in manuscript.

[Figure]

Fig. iiii Chemical domains of aerosol response to ammonia and nitrate emissions at 295K and 298K

*5. The authors considered that ALW equals to 75 is the determining parameter for haze generations in studied area, however, for which step in the haze processes is equivalent to ALW=75. The definition of the haze processes is far easier than the definition of ALWC.*

**Response:** Thanks a lot. The ALWC of each pollution stages were supplied in Table S1 in supplement accordingly, in which the ALWC of Moderate pollution stage was equivalent to 75 μg/m³.

*6. Line 244, it should not be the pH of secondary inorganic aerosol.*
**Response:** Thanks. It had been revised as "pH of aerosol" in the revision.

*7. Line 287, I can hardly find Fig.S9a corresponded to the S-curve proposed by Guo et al. (2017). Pls. check carefully.*
**Response:** Thanks a lot. Fig. S9 had three sub-graphs, and the fitted S-curve were shown as the solid blue line in Fig. S9b. Accordingly, this sentence had been revised.

*8. Line 362-365, grammar check.*
**Response:** Thanks. This sentence had been rewritten according to your comments.

*9. The details of solar spectrophotometer did not mention in the manuscript. pls. provide it. In addition, Line 314-316 was poorly expressed, serious pollution stages were much higher than 1/4.*

**Response:** Thanks for your comments. The details of solar spectrophotometer were updated in the revision (section 2.2.1). In addition, the sentence (Line 314-316 of original manuscript) had been revised to avoid the possible misunderstanding.

*10. Detailed calculation method of aqueous $NO_3^-$ concentrations should place in this manuscript to make easier reading for readers.*

**Response:** Thank you for your suggestions. The method of aqueous $NO_3^-$ concentrations had been provided in section S1 of Supplement.

*11. Compared with the sulfate heterogeneous rate calculation, NOR, SOR and NTR used more frequently in this manuscript and were the key parameters on mechanism presentation, thus, I recommend authors a fully rewritten on method part after a fully consideration.*

**Response:** Thank you for your suggestion. Accordingly, the method part had been rewritten and the detailed calculations of NOR, SOR and NTR had been moved from Supplement to the main text.

**Reference:**

Atkinson, R., Baulch, D. L., Cox, R. A., Jr., R. F. H., Kerr, J. A., Rossi, M. J., and Troe, J.: Evaluated Kinetic, Photochemical and Heterogeneous Data for Atmospheric Chemistry: Supplement V. IUPAC Subcommittee on Gas Kinetic Data Evaluation for Atmospheric Chemistry, J. Phys. Chem. Ref. Data, 26, 521-1011, 10.1063/1.556011, 1997.

Guo, S., Hu, M., Zamora, M. L., Peng, J., Shang, D., Zheng, J., Du, Z., Wu, Z., Shao, M., Zeng, L., Molina, M. J., and Zhang, R.: Elucidating severe urban haze formation in China, Proc. Natl. Acad. Sci., 111, 17373-17378, 10.1073/pnas.1419604111, 2014.

He, P., Alexander, B., Geng, L., Chi, X., Fan, S., Zhan, H., Kang, H., Zheng, G., Cheng, Y., Su, H., Liu, C., and Xie, Z.: Isotopic constraints on heterogeneous sulfate production in Beijing haze, Atmos. Chem. Phys., 18, 5515-5528, 10.5194/acp-18-5515-2018, 2018.

Huang, X., Qiu, R., Chan, C. K., and Ravi Kant, P.: Evidence of high $PM_{2.5}$ strong acidity in ammonia-rich atmosphere of Guangzhou, China: Transition in pathways of ambient ammonia to form aerosol ammonium at $[NH_4^+]/[SO_4^{2-}]$ =1.5, Atmos. Res., 99, 488-495, https://doi.org/10.1016/j.atmosres.2010.11.021, 2011.

Jia, S., Chen, W., Zhang, Q., Krishnan, P., Mao, J., Zhong, B., Huang, M., Fan, Q., Zhang, J., Chang, M., Yang, L., and Wang, X.: A quantitative analysis of the driving factors affecting seasonal variation of aerosol pH in Guangzhou, China, Sci. Total Environ., 725, 138228, https://doi.org/10.1016/j.scitotenv.2020.138228, 2020.

Jiang, B. and Xia, D.: Role identification of $NH_3$ in atmospheric secondary new particle formation in haze occurrence of China, Atmos. Environ., 163, 107-117, https://doi.org/10.1016/j.atmosenv.2017.05.035, 2017.

Li, L., Duan, Z., Li, H., Zhu, C., Henkelman, G., Francisco, J. S., and Zeng, X. C.: Formation of HONO from the $NH_3$-promoted hydrolysis of $NO_2$ dimers in the atmosphere, Proc. Natl. Acad. Sci., 115, 7236-7241, doi:10.1073/pnas.1807719115, 2018.

Liu, Z., Xie, Y., Hu, B., Wen, T., Xin, J., Li, X., and Wang, Y.: Size-resolved aerosol water-soluble ions during the summer and winter seasons in Beijing: Formation mechanisms of secondary inorganic aerosols, Chemosphere, 183, 119-131, https://doi.org/10.1016/j.chemosphere.2017.05.095, 2017.

Malloy, Q. G. J., Li, Q., Warren, B., Cocker Iii, D. R., Erupe, M. E., and Silva, P. J.: Secondary organic aerosol formation from primary aliphatic amines with $NO_3$ radical, Atmos. Chem. Phys., 9, 2051-2060, 10.5194/acp-9-2051-2009, 2009

Mozurkewich, M.: The dissociation constant of ammonium nitr.ate and its dependence on temperature, relative humidity and particle size, Atmos. Environ., Part A. General Topics, 27, 261-270, https://doi.org/10.1016/0960-1686(93)90356-4, 1993.

Nenes, A., Pandis, S. N., Weber, R. J., and Russell, A.: Aerosol pH and liquid water content determine when particulate matter is sensitive to ammonia and nitrate availability, Atmos. Chem. Phys., 20, 3249-3258, 10.5194/acp-20-3249-2020, 2020.

Pathak, R. K., Wu, W. S., and Wang, T.: Summertime PM2.5 ionic species in four major cities of China: nitrate formation in an ammonia-deficient atmosphere, Atmos. Chem. Phys., 9, 1711-1722, 10.5194/acp-9-1711-2009, 2009.

Pathak, R. K., Yao, X., and Chan, C. K.: Sampling Artifacts of Acidity and Ionic Species in $PM_{2.5}$, Environ. Sci. Technol., 38, 254-259, 10.1021/es0342244, 2004.

Seinfeld, J. H., Pandis, S. N., and Noone, K. J.: Atmospheric Chemistry and Physics: From Air Pollution to Climate Change, Phys. Today, 51, 88-90, 1997.

Shao, J., Chen, Q., Wang, Y., Lu, X., He, P., Sun, Y., Shah, V., Martin, R. V., Philip, S., Song, S., Zhao, Y., Xie, Z., Zhang, L., and Alexander, B.: Heterogeneous sulfate aerosol formation mechanisms during wintertime Chinese haze events: air quality model assessment using observations of sulfate oxygen isotopes in Beijing, Atmos. Chem. Phys., 19, 6107-6123, 10.5194/acp-19-6107-2019, 2019.

Sharma, B., Jia, S., Polana, A. J., Ahmed, M. S., Haque, R. R., Singh, S., Mao, J., and Sarkar, S.: Seasonal variations in aerosol acidity and its driving factors in the eastern Indo-Gangetic Plain: A quantitative analysis, Chemosphere, 305, 135490, https://doi.org/10.1016/j.chemosphere.2022.135490, 2022.

Shi, G., Xu, J., Shi, X., Liu, B., Bi, X., Xiao, Z., Chen, K., Wen, J., Dong, S., Tian, Y., Feng, Y., Yu, H., Song, S., Zhao, Q., Gao, J., and Russell, A. G.: Aerosol pH Dynamics During Haze Periods in an Urban Environment in China: Use of Detailed, Hourly, Speciated Observations to Study the Role of Ammonia Availability and Secondary Aerosol Formation and Urban Environment, J. Geophys. Res.: Atmos., 124, 9730-9742, https://doi.org/10.1029/2018JD029976, 2019.

Tan, H., Cai, M., Fan, Q., Liu, L., Li, F., Chan, P. W., Deng, X., and Wu, D.: An analysis of aerosol liquid water content and related impact factors in Pearl River Delta, Sci. Total Environ., 579, 1822-1830, https://doi.org/10.1016/j.scitotenv.2016.11.167, 2017.

Tang, X., Zhang, Y., and Shao, M.: Atmospheric Environmental Chemistry, Higher Education Press, Beijing, 2018.

Tao, Y., Ye, X., Ma, Z., Xie, Y., Wang, R., Chen, J., Yang, X., and Jiang, S.: Insights into different nitrate formation mechanisms from seasonal variations of secondary inorganic aerosols in Shanghai, Atmos. Environ., 145, 1-9, https://doi.org/10.1016/j.atmosenv.2016.09.012, 2016.

Wang, G. H., Cheng, C. L., Huang, Y., Tao, J., Ren, Y. Q., Wu, F., Meng, J. J., Li, J. J., Cheng, Y. T.,

Cao, J. J., Liu, S. X., Zhang, T., Zhang, R., and Chen, Y. B.: Evolution of aerosol chemistry in Xi'an, inland China, during the dust storm period of 2013 - Part 1: Sources, chemical forms and formation mechanisms of nitrate and sulfate, Atmos. Chem. Phys., 14, 11571-11585, 10.5194/acp-14-11571-2014, 2014.

Wang, H., Wang, X., Zhou, H., Ma, H., Xie, F., Zhou, X., Fan, Q., Lü, C., and He, J.: Stoichiometric characteristics and economic implications of water-soluble ions in $PM_{2.5}$ from a resource-dependent city, Environ. Res., 193, 110522, https://doi.org/10.1016/j.envres.2020.110522, 2021.

Wang, S., Nan, J., Shi, C., Fu, Q., Gao, S., Wang, D., Cui, H., Saiz-Lopez, A., and Zhou, B.: Atmospheric ammonia and its impacts on regional air quality over the megacity of Shanghai, China, Sci. Rep-UK, 5, 15842, 10.1038/srep15842, 2015.

Wang, S., Yin, S., Zhang, R., Yang, L., Zhao, Q., Zhang, L., Yan, Q., Jiang, N., and Tang, X.: Insight into the formation of secondary inorganic aerosol based on high-time-resolution data during haze episodes and snowfall periods in Zhengzhou, China, Sci. Total Environ., 660, 47-56, https://doi.org/10.1016/j.scitotenv.2018.12.465, 2019.

Zheng, B., Zhang, Q., Zhang, Y., He, K. B., Wang, K., Zheng, G. J., Duan, F. K., Ma, Y. L., and Kimoto, T.: Heterogeneous chemistry: a mechanism missing in current models to explain secondary inorganic aerosol formation during the January 2013 haze episode in North China, Atmos. Chem. Phys., 15, 2031-2049, 10.5194/acp-15-2031-2015, 2015.

Zhou, H., Lü, C., He, J., Gao, M., Zhao, B., Ren, L., Zhang, L., Fan, Q., Liu, T., He, Z., Dudagula, Zhou, B., Liu, H., and Zhang, Y.: Stoichiometry of water-soluble ions in $PM_{2.5}$: Application in source apportionment for a typical industrial city in semi-arid region, Northwest China, Atmos. Res., 204, 149-160, https://doi.org/10.1016/j.atmosres.2018.01.017, 2018.

---

## Author Response (AR2)

**Response to the reviewers**

Dear reviewers and editors,

Our manuscript entitled "The shifting of secondary inorganic aerosols formation mechanism during haze aggravation: The decisive role of aerosol liquid water (acp-2022-590)" has been revised according to the comments raised by the reviewers. We are very thankful the comments from the reviewers and editors which largely improved the quality of our manuscript. The changes have been marked in the track-changes file. The detailed and point-by-point response to the reviewer comments were detailed below.

***Reviewer 2[#]:***

I think the manuscript is much better than before. However, I'm still not sure why the ratios ($PM_{1.0}/PM_{2.5}$ and $PM_{2.5}/PM_{10}$) can be used as the proxy for aerosol hygroscopicity. Are there any previous studies that found this conclusion or that this strong conclusion was promoted by the authors? I agree with the authors that the contribution of accumulation mode particles to ALWC dominated among all the aerosol particle modes. However, the hygroscopicity in $PM_{1.0}$ itself will vary significantly under different environments. They observed significant correlations of ALWC with the ratios ($PM_{1.0}/PM_{2.5}$ and $PM_{2.5}/PM_{10}$) in this work. The reason, I guess, is that the secondary aerosol formation mainly happens on these $PM_{1.0}$ particles as the surface area and volume of the $PM_{1.0}$ particles are much larger than those of the coarse particles.

**Response:** Thank you again for your suggestions. Previous works reported aerosol hygroscopicity is typically quantified by the hygroscopic growth factor (HGF) or hygroscopicity parameter ($\kappa$). When the particle size increases from 50 nm to several hundreds of nanometers during haze development, the values of HGF and $\kappa$ typically increase from 1.2 and 0.15 to 1.5 and 0.3, respectively, significantly correlating with inorganic fractions. The increase in aerosol hygroscopicity will result in the acceleration of aqueous chemistry processes and the haze aggravation. We agree with you that secondary aerosol formation mainly happens on these $PM_{1.0}$ particles as the surface area and volume of the $PM_{1.0}$ particles are much larger than those of the coarse particles. Thus, the larger surface area of $PM_{1.0}$ inherently promotes the partitioning of more inorganic fractions on particles. As a result of the co-effects of thermodynamic stability and higher RH, more ALW is required and presents positive feedback toward more

components partitioning on the particulates, which in turn promotes the gradual growth of smaller particle sizes and the haze aggravation. Accordingly, the ratios of $PM_{1.0}/PM_{2.5}$ and $PM_{2.5}/PM_{10}$ presented the potential possibility to index the hygroscopic growth of particulate matter. We greatly appreciate your professional suggestion, which helps us to better explain the results in theory. In the revision, this section has been revised in light of your comments.

References

Guo, S.; Hu, M.; Zamora, M. L.; Peng, J. F.; Shang, D. J.; Zheng, J.; Du, Z. F.; Wu, Z.; Shao, M.; Zeng, L. M.; Molina, M. J.; Zhang, R. Y. Elucidating severe urban haze formation in China. Proc. Natl. Acad. Sci. U. S. A. 2014, 111 (49), 17373−17378

Wu, Z. J.; Zheng, J.; Shang, D. J.; Du, Z. F.; Wu, Y. S.; Zeng, L. M.; Wiedensohler, A.; Hu, M. Particle hygroscopicity and its link to chemical composition in the urban atmosphere of Beijing, China, during summertime. Atmos. Chem. Phys. 2016, 16 (2), 1123−1138

Liu, Y. C.; Wu, Z. J.; Wang, Y.; Xiao, Y.; Gu, F. T.; Zheng, J.; Tan, T. Y.; Shang, D. J.; Wu, Y. S.; Zeng, L. M.; Hu, M.; Bateman, A. P.; Martin, S. T. Submicrometer Particles Are in the Liquid State during Heavy Haze Episodes in the Urban Atmosphere of Beijing, China. Environ. Sci. Technol. Lett. 2017, 4 (10), 427−432

Tie, X. X.; Huang, R. J.; Cao, J. J.; Zhang, Q.; Cheng, Y. F.; Su, H.; Chang, D.; Poschl, U.; Hoffmann, T.; Dusek, U.; Li, G. H.; Worsnop, D. R.; O'Dowd, C. D. Severe Pollution in China Amplified by Atmospheric Moisture. Sci. Rep. 2017, 7, 15760